# Diffusion Transformers as Open-World Spatiotemporal Foundation Models

**Yuan Yuan[1], Chonghua Han[1], Jingtao Ding[1], Guozhen Zhang[2], Depeng Jin[1], Yong Li[1,*]**
[1] Department of Electronic Engineering, BNRist, Tsinghua University
[2] TsingRoc.ai
Beijing, China
[*]Corresponding author: liyong07@tsinghua.edu.cn

## Abstract

The urban environment is characterized by complex spatio-temporal dynamics arising from diverse human activities and interactions. Effectively modeling these dynamics is essential for understanding and optimizing urban systems. In this work, we introduce UrbanDiT, a foundation model for open-world urban spatio-temporal learning that successfully scales up diffusion transformers in this field. UrbanDiT pioneers a unified model that integrates diverse data sources and types while learning universal spatio-temporal patterns across different cities and scenarios. This allows the model to unify both multi-data and multi-task learning, and effectively support a wide range of spatio-temporal applications. Its key innovation lies in the elaborated prompt learning framework, which adaptively generates both data-driven and task-specific prompts, guiding the model to deliver superior performance across various urban applications. UrbanDiT offers three advantages: 1) It unifies diverse data types, such as grid-based and graph-based data, into a sequential format; 2) With task-specific prompts, it supports a wide range of tasks, including bi-directional spatio-temporal prediction, temporal interpolation, spatial extrapolation, and spatio-temporal imputation; and 3) It generalizes effectively to open-world scenarios, with its powerful zero-shot capabilities outperforming nearly all baselines with training data. UrbanDiT sets up a new benchmark for foundation models in the urban spatio-temporal domain. Code and datasets are publicly available at `https://github.com/tsinghua-fib-lab/UrbanDiT`.

## 1 Introduction

The urban environment is characterized by complex spatio-temporal dynamics arising from diverse human activities and interactions within the city. These dynamics are reflected in different types of data. For example, grid-based data divides urban space into regular cells, often used to track crowd flows. In contrast, graph-based data represents spatial structures like road networks as nodes and edges, such as traffic speeds on roads. The data from different cities are usually with unique layouts, infrastructures, and planning strategies. Effectively modeling their spatio-temporal dynamics is crucial for optimizing urban services and understanding how cities function. Therefore, it raises an essential research question: can we develop a foundation model, similar to those in natural language processing [41, 4] and computer vision [3, 26, 10], that learns universal spatio-temporal patterns and serves as a general-purpose model for various urban applications?

In the context of urban spatio-temporal modeling, recent advancements such as GPD [53], UrbanGPT [23], and UniST [51] have opened exciting avenues for understanding complex urban dynamics. As compared in Table 1, these models either utilize LLMs [23] or develop unified models from scratch [51, 53] tailored for urban spatio-temporal predictions. By training on multiple datasets,

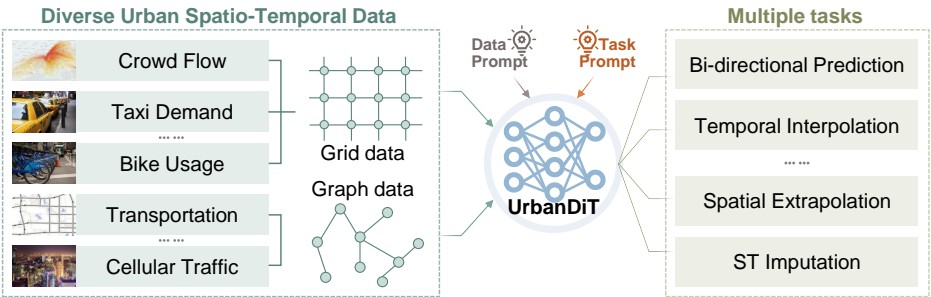

Figure 1: A diagram of UrbanDiT: a foundation model that integrates diverse data sources while addressing multiple tasks.

they have showcased impressive generalization capabilities. However, their focus remains largely on prediction tasks, and they are often restricted to specific data types—such as grid-based data [23, 51] or graph-based traffic data [53]. Thus, realizing the full potential of foundation models capable of seamlessly handling diverse data types, sources, and tasks in open-world scenarios remains an open and largely unexplored area of research.

Urban spatio-temporal data is usually defined by varying spatio-temporal resolutions and complex interactions among entities. Building a foundation model requires a scalable architecture capable of accommodating these complexities. Moreover, the intricate nature of urban spatio-temporal dynamics necessitates a model that can learn from complex data distributions. Diffusion Transformers, exemplified by models like Sora [3], offer a compelling solution for this purpose. By combining the generative power of diffusion processes with the scalability and flexibility of transformer architectures, diffusion transformers present a promising backbone.

In this work, we introduce UrbanDiT, which unifies training across diverse urban scenarios and tasks, effectively scaling up diffusion transformers for comprehensive urban spatio-temporal learning. It offers three appealing benefits: 1) It unifies diverse data types into a sequential format, allowing it to capture spatio-temporal patterns across various cities and domains. 2) It supports a wide range of tasks with a single model via task-specific prompts, without the need for re-training across different tasks. 3) It generalizes well to open-world scenarios, exhibiting powerful zero-shot performance. To build UrbanDiT, we first unify different input data by converting it into the sequential format. We transformer blocks as the denoising network, which are equipped with both temporal and spatial attention modules. To integrate diverse data types and tasks, we propose a unified prompt learning framework. It maintains memory pools to capture learned spatio-temporal patterns and generate data-driven prompts, while also create task-specific prompts for various spatio-temporal tasks. These prompts are concatenated into the unified sequential input before being fed into the transformer modules. The design of prompt learning serves as a flexible intermediary, adaptable to a wide range of scenarios.

UrbanDiT, built on the DiT backbone with a prompt learning framework, is a pioneering open-world foundation model. It excels at handling diverse urban spatio-temporal data and a wide range of tasks, including bi-directional spatio-temporal prediction, temporal interpolation, spatial extrapolation, and spatio-temporal imputation. This makes UrbanDiT a powerful and universal solution for various urban spatio-temporal applications. We summarize our contributions as follows:

- To the best of our knowledge, we are the first to explore a foundation model for general-purpose urban spatio-temporal learning, integrating diverse data types and multiple urabn tasks within a single unified model.

- We present UrbanDiT, an open-world foundation model built on diffusion transformers. Through our proposed prompt learning, UrbanDiT effectively brings together heterogeneous spatio-temporal data and tasks, using data-driven and task-specific prompts to enhance performance.

- Extensive experiments demonstrate that UrbanDiT effectively captures complex urban spatio-temporal dynamics, achieving state-of-the-art performance across multiple datasets and tasks. It also exhibits powerful zero-shot capabilities, proving its applicability in open-world settings. UrbanDiT marks a significant step forward in the advancement of urban foundation models.

Table 1: Comparison between existing models and UrbanDiT across five aspects.

| Method | Model Init. | Data Type | Diverse Data Sources | Task Flexibility | Zero-shot |
|---|---|---|---|---|---|
| GPD [53] | Scratch | Graph | × | × | × |
| UniST [51] | Scratch | Grid | ✓ | × | ✓ |
| UrbanGPT [23] | LLMs | Grid | ✓ | × | ✓ |
| CityGPT [12] | LLMs | Languages | × | ✓ | × |
| UrbanDiT | Scratch | Graph/Grid | ✓ | ✓ | ✓ |

## 2 Related Work

**Urban Spatio-Temporal Learning.** Urban spatio-temporal learning encompasses a variety of tasks such as prediction [39, 2, 52, 22, 54], interpolation [1, 16], extrapolation [30, 29], and imputation [40, 18], addressing how urban systems evolve across space and time. Deep learning has achieved significant progress in these areas, with techniques ranging from CNNs [22, 54], RNNs [43, 42, 24], MLPs [36], GNNs [2, 15], and Transformers [7, 19], to the more recent use of diffusion models [52, 40, 45]. Each of these approaches has been employed to model complicated spatio-temporal relationships inherent to urban environments. However, most existing models are tailored to specific datasets and tasks. In contrast, our approach is designed to handle multiple tasks and generalize across diverse urban scenarios.

**Urban Foundation Models.** Foundation models have made significant progress in language models [11, 41, 4] and image generation [3, 26, 10]. Recently, researchers have extended the concept of foundation models to urban environments, aiming to address unique challenges of urban spatio-temporal data. Some representative works in this area include UrbanGPT [23], UniST [51], and CityGPT [13]. UrbanGPT introduces LLMs designed for spatio-temporal predictions within urban contexts. UniST develops a foundation model from scratch specifically for urban prediction tasks, demonstrating zero-shot capabilities that allow the model to generalize to new scenarios without additional training. CityGPT, on the other hand, focuses on enhancing the LLM's ability to comprehend and solve urban tasks by improving its understanding of urban spaces. Table 1 provides a comparison of key abilities across existing urban foundation models and UrbanDiT. As shown, UrbanDiT is trained from scratch, allowing it to fully leverage data diversity while offering flexibility across a wide range of tasks. Compared to previous efforts, UrbanDiT represents a significant advancement in developing urban foundation models.

**Diffusion Models for Spatio-Temporal Data.** Diffusion models, originally popularized in image generation, have recently gained attention in handling spatio-temporal data and time series. They iteratively add and remove noise from data, allowing them to capture complex patterns across both temporal and spatial dimensions [49, 52, 18, 45, 34]. In the context of time series, diffusion models have been applied to tasks such as forecasting [21, 34] and imputation [48, 40], outperforming traditional methods by generating more accurate and coherent sequences. For spatio-temporal data, diffusion models have proven useful in a variety of tasks, including traffic prediction [45], environmental monitoring [52], and human mobility generation [60, 59]. By effectively modeling spatio-temporal dependencies, these models can capture both the spatial correlations and temporal dynamics inherent in urban systems. UrbanDiT leverages the generative power of diffusion models to capture complex urban spatio-temporal patterns, while its flexible conditioning mechanisms allow it to address a wide range of spatio-temporal tasks.

## 3 Method

### 3.1 Preliminary

**Urban Spatio-Temporal Data.** Urban spatio-temporal data typically falls into two categories: *grid-based* and *graph-based data*. Grid-based data is structured in a uniform grid layout. Graph-based data, on the other hand, highlights connectivity, capturing the relationships between various urban entities like streets and intersections. For both different spatial organizations, the temporal dimension is characterized as time series data. The data can be denoted as $X^{N \times T}$, where $N$ denotes the number of spatial partitions. For graph-based data, $N$ corresponds to the number of nodes, while for grid-based data, it is defined as the product of the height and width of the grid ($N = H \times W$). This enables a unified representation of urban spatio-temporal data with different spatial organizations.

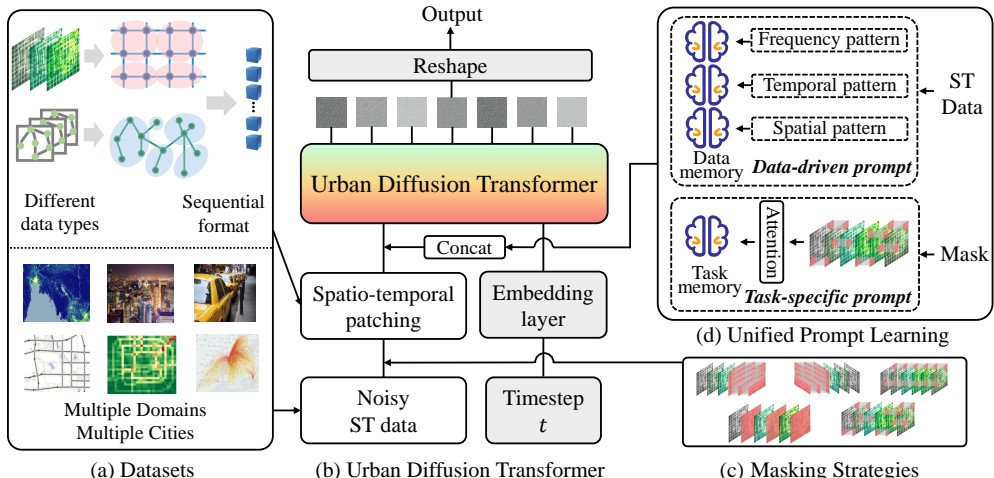

Figure 2: Illustration of the whole framework of UrbanDiT, including four key components: a) Unifying different urban spatio-temporal data types; b) The diffusion pipeline of our UrbanDiT; c) Different masking strategies to specify different tasks; d) Unified prompt learning with data-driven and task-specific prompts to enhance the denoising process.

**Urban Spatio-Temporal Tasks.** In addition to the commonly recognized (1) *forward prediction* task, urban spatio-temporal analysis encompasses several other critical tasks. (2) *Backward Prediction* involves estimating past states based on current or future data. It is essential for understanding historical trends and validating predictive models. (3) *Temporal Interpolation* aims to estimate values at unobserved time points within a known temporal range. (4) *Spatial Extrapolation* involves predicting values beyond the observed spatial domain. (5) *Spatio-Temporal Imputation* refers to the process of filling in missing values in spatio-temporal datasets.

### 3.2 UrbanDiT

Figure 2 illustrates the overall framework of UrbanDiT, which is based on diffusion transformers. This framework seamlessly integrates various data types and tasks into a cohesive model.

**Unification of Data and Tasks.** We convert data, characterized by a three-dimensional structure (2D spatial and 1D temporal dimensions), into a unified sequential format. For the temporal dimension, we employ patching techniques commonly used in foundational models for time series [32]. For grid-based data, we apply 2D patching methods, which are widely utilized in image processing, to organize the data. This allows us to rearrange the three-dimensional grid data into a one-dimensional sequential format. For graph-based data, we use Graph Convolutional Networks (GCN) [55] to process each node and integrate it with the temporal dimension to reshape the data into a one-dimensional format as well. More details of data unification can be found in Appendix B.1

To adapt to various tasks, we employ a unified masking strategy. These tasks can be framed as reconstructing missing parts of the data, with distinct masking strategies tailored to each task. For *Forward Prediction*, we mask future time steps while utilizing past and present data points to predict the missing values. Conversely, for *Backward Prediction*, we mask past time steps to estimate historical values based on current and future observations. In the case of *temporal interpolation* tasks, we apply masks to specific time points within a continuous series, allowing the model to fill in these gaps. For *spatio-temporal imputation*, we randomly mask missing values across both spatial and temporal dimensions, enabling the model to leverage surrounding context for accurate estimations. Finally, in *spatial extrapolation* tasks, we mask areas outside the observed spatial domain to predict values for unobserved regions based on existing spatial patterns. Consequently, the input of the denoising network $X^t$ is represented as the concatenation of noise features and unmasked spatio-temporal data (conditional observations):

$$X^t = X^t * (1 - M) + X^0 * M$$

where $X^t$ denotes the noise features, $M$ is the mask that controls the availability of values for downstream tasks, and $X^0$ represents the clean values of the spatio-temporal data. In this way, we can modulate different masks $M$ to facilitate various urban spatio-temporal applications.

**Sequential Input of Spatio-Temporal Data.** We first apply temporal patching to process time series data at each spatial location, represented as $X^{N \times T' \times D} = \text{CONV}(X^{N \times T \times D})$, where $T' = \frac{T}{p_t}$ and $p_t$ is the temporal patch size. Next, for grid-based data, we implement 2D spatial patching, resulting in $X_p = \text{CONV}_{2D}(X^{H \times W \times T' \times D})$, where $X_p \in \mathbb{R}^{L \times D}, L = \frac{H \times W \times T}{p_s \times p_s \times p_t}$. In this way, we effectively reorganize the data into a format well-suited for transformers.

**Spatio-Temporal Transformer Block.** The overall model is composed of multiple spatio-temporal transformer blocks. Each block employs both temporal attention and spatial attention, with spatial and temporal attention mechanisms operating independently. This design choice is made to enhance computational efficiency, as the complexity of attention scales with the square of the sequence length.

**Diffusion Transformer.** We adopt the diffusion transformer model, which integrates a denoising network designed to process complex inputs effectively. The inputs to the denoising network consist of three key components: the noisy spatio-temporal data, the timestep, and the prompt. For the timestep $t$, we utilize them for layer normalization following previous practices [33, 28], which helps stabilize and standardize the input features at each timestep. The prompt, which provides contextual information or guidance for the model, is concatenated with the input data to enhance the model's understanding of the data and task at hand. This concatenation is straightforward due to the transformer's capability to manage variable sequence lengths, providing flexibility in processing diverse inputs. By incorporating these elements, the diffusion transformer model effectively learns to denoise and generate robust desired results in spatio-temporal contexts.

### 3.3 Unified Prompt Learning

**Data-Driven Prompt.** The data-driven prompt is crucial for training a unified model with multiple and diverse datasets, as such datasets often exhibit significant variations in patterns and distributions. In this context, the prompt acts as a guiding mechanism, helping the model to effectively navigate these differences and generate accurate results. Similar to retrieval-augmented generation, prompts retrieve the most relevant information, enhancing the model's ability to contextualize and interpret spatio-temporal data. By aligning the model's learning process with the specific characteristics of various spatio-temporal patterns, data prompts ensure that UrbanDiT can adaptively respond to a wide range of urban spatio-temporal scenarios.

To achieve this goal, we employ memory networks, specifically utilizing three memory pools designed to capture the time-domain, frequency-domain and spatial patterns of spatio-temporal dynamics. For different input data, the prompt network retrieves prompts from these memory pools based on the respective time-domain, frequency-domain, and spatial patterns. As shown in Figure 3, each memory pool is structured as a key-value store $(K_t, V_t) = \{(k_t^1, v_t^1), ..., (k_t^N, v_t^N)\}$, $(K_f, V_f) = \{(k_f^1, v_f^1), ..., (k_f^N, v_f^N)\}$, $(K_s, V_s) = \{(k_s^1, v_s^1), ..., (k_s^N, v_s^N)\}$, where both keys and values are learnable embeddings and randomly initialized. The data-driven prompts are generated as follows:

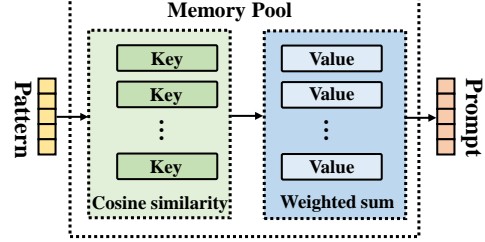

Figure 3: Structure of memory pools.

$$\alpha_t = \text{SOFTMAX}(X_t, K_t), \quad P_t = \sum \alpha_t \cdot V_t,$$
$$\alpha_f = \text{SOFTMAX}(X_f, K_f), \quad P_f = \sum \alpha_f \cdot V_f,$$
$$\alpha_f = \text{SOFTMAX}(X_s, K_s), \quad P_s = \sum \alpha_s \cdot V_s,$$
$$X = \text{CONCAT}(P_t, P_f, X).$$

**Task-Specific Prompt.** We design task-specific prompts to enhance the model's performance across different tasks. These prompts are generated from the mask, and we employ attention mechanisms

to obtain the mask prompt $P_m$ from the mask map as $P_m = \text{ATTENTION}(\text{FLATTEN}(M))$. The learned pattern $P_m$ is then concatenated with the input sequence, resulting in $X = \text{CONCAT}(P_m, X)$. This enables the model to effectively incorporate task-specific information. We provide details of data-driven and task-specific prompts in Appendix B.2

### 3.4 Training and Inference

The training process alternates between multiple datasets and tasks. In each iteration, we randomly select a dataset and a corresponding task to perform gradient descent training. This approach enhances the model's robustness by exposing it to diverse scenarios and helps prevent overfitting by ensuring the model learns from a wide range of inputs and objectives. Let $D = \{D_1, D_2, \ldots, D_m\}$ represent the set of datasets, and $T = \{T_1, T_2, \ldots, T_k\}$ denote the set of tasks. Let $\mathcal{L}(d_i, t_i)$ be the loss function for the chosen dataset $d_i$ and task $t_i$, with the model parameters denoted as $\theta$. Overall, the training process is summarized as follows:

$$\text{For } i = 1 \text{ to } N: \quad d_i \sim \text{Uniform}(D), \quad t_i \sim \text{Uniform}(T)$$
$$\Rightarrow \quad \theta \leftarrow \theta - \eta \nabla \mathcal{L}(d_i, t_i; \theta)$$

where $N$ is the total number of training iterations and $\eta$ is the learning rate.

For the training of the UrbanDiT model, we adopt a novel diffusion training approach proposed by the InstaFlow [26], which significantly improves the efficiency of spatio-temporal data generation. By employing rectified flow, it is an ordinary differential equation (ODE)-based framework that aligns the noise and data distributions through a straightened trajectory, as opposed to the curved paths often seen in traditional models.

## 4 Performance Evaluations

**Datasets.** We utilize a diverse set of datasets from multiple domains and cities to evaluate urban spatio-temporal applications, which include taxi demand, cellular network traffic, crowd flows, transportation traffic, and dynamic population, reflecting a broad spectrum of urban activities. The datasets are sourced from different cities such as New York City, Beijing, Shanghai, and Nanjing, each representing unique urban characteristics. These datasets vary significantly in their spatial structures (e.g., grid or graph formats), the number of locations, and their spatial and temporal resolutions. These variations are influenced by differences in city structures, urban planning strategies, and data collection methodologies across regions. For a detailed summary of the datasets, please refer to Table 4 and Table 5 in Appendix A. We split the datasets into training, validation, and testing sets along the temporal dimension, using a 6:2:2 ratio. To ensure no overlap between them, we carefully remove any overlapping points, ensuring clear separation across the temporal splits for evaluation.

**Baselines.** To evaluate the performance of UrbanDiT, we establish a comprehensive benchmark, comparing it against state-of-the-art models across different urban tasks. For prediction tasks, we include both traditional time series models such as Historical Average (HA) and ARIMA, as well as advanced deep learning-based spatio-temporal models like STResNet [54], ACFM [25], STNorm [9], STGSP [58], MC-STL [56], PromptST [57], STID [36], and UniST [51]. Additionally, we compare against leading video prediction models, including SimVP [14], TAU [38], MAU [6], and MIM [44], as well as recent time series forecasting approaches such as PatchTST [32], iTransformer [27], Time-LLM [20], and the diffusion-based model CSDI [40]. For graph-based datasets, we evaluate UrbanDiT against cutting-edge spatio-temporal graph models, including STGCN [50], DCRNN [22], GWN [47], MTGNN [46], AGCRN [2], GTS [35], and STEP [37]. Furthermore, for spatio-temporal imputation tasks, we compare our model with state-of-the-art baselines such as CSDI, ImputeFormer [31], Grin [8], and BriTS [5], adapting these methods for temporal interpolation and spatial extrapolation tasks. We provide more details of baselines in Appendix C.1

### 4.1 Comparison to the State-of-the-art

**Bi-directional Spatio-Temporal Prediction.** For this task, we set both the historical input window and prediction horizon to 12 time steps. Depending on the dataset, the temporal granularity varies—12 steps may correspond to 1 hour for datasets with 5-minute intervals, 6 hours for datasets with 30-minute intervals, and 12 hours for those with 1-hour intervals. For baselines that cannot handle

| | TaxiBJ | | FlowSH | | TaxiNYC | | CrowdNJ | | PopBJ | |
|---|---|---|---|---|---|---|---|---|---|---|
| **Model** | **MAE** | **RMSE** | **MAE** | **RMSE** | **MAE** | **RMSE** | **MAE** | **RMSE** | **MAE** | **RMSE** |
| HA | 53.03 | 91.55 | 13.43 | 38.92 | 26.49 | 77.10 | 0.48 | 0.93 | 0.232 | 0.343 |
| ARIMA | 57.5 | 291 | 9.15 | 26.70 | 23.91 | 99.22 | 0.443 | 0.989 | 0.236 | 0.404 |
| STResNet | 26.55 | 37.96 | 45.63 | 59.82 | 14.81 | 26.88 | 0.511 | 0.718 | 0.546 | 0.751 |
| ACFM | 19.87 | 30.95 | 24.95 | 46.92 | 9.85 | 20.82 | 0.284 | 0.468 | 0.141 | 0.200 |
| STNorm | 19.00 | 31.21 | 11.88 | 28.46 | 10.43 | 26.94 | 0.231 | 0.384 | 0.132 | 0.198 |
| STGSP | 17.54 | 27.31 | 17.54 | 38.77 | 10.52 | 25.94 | 0.263 | 0.410 | 0.157 | 0.229 |
| MC-STL | 28.51 | 38.50 | 33.83 | 46.06 | 26.01 | 36.75 | 0.727 | 0.504 | 0.235 | 0.311 |
| MAU | 46.37 | 71.07 | 21.38 | 45.04 | 21.79 | 49.15 | 0.402 | 0.648 | 0.166 | 0.256 |
| MIM | 42.40 | 68.18 | 22.49 | 47.29 | 9.151 | 24.53 | 0.399 | 0.715 | 0.214 | 0.298 |
| SimVP | 21.67 | 35.58 | 15.87 | 28.59 | 9.08 | 19.69 | 0.191 | 0.282 | 0.148 | 0.213 |
| TAU | 15.86 | 26.43 | 15.22 | 26.04 | 9.08 | 19.46 | 0.219 | 0.326 | 0.135 | 0.196 |
| PromptST | 16.12 | 27.42 | 9.37 | 23.01 | 8.24 | 22.82 | 0.161 | 0.306 | 0.099 | 0.171 |
| UniST | 14.04 | 23.67 | 9.10 | 19.95 | 5.85 | 17.55 | 0.119 | 0.191 | 0.106 | 0.172 |
| STID | 16.36 | 25.55 | 12.92 | 21.19 | 8.32 | 18.49 | 0.160 | 0.234 | 0.203 | 0.262 |
| PatchTST | 30.55 | 53.36 | 10.69 | 28.17 | 17.03 | 50.45 | 0.223 | 0.465 | 0.189 | 0.291 |
| PatchTST-all | 33.62 | 60.55 | 12.16 | 31.79 | 21.27 | 58.61 | 0.403 | 0.811 | 0.176 | 0.279 |
| iTransformer | 24.05 | 42.17 | 10.19 | 25.91 | 45.19 | 45.19 | 0.216 | 0.466 | 0.154 | 0.249 |
| Time-LLM | 29.55 | 51.20 | 10.57 | 28.19 | 17.65 | 52.94 | 0.210 | 0.405 | 0.115 | 0.195 |
| CSDI | 14.76 | 25.87 | 8.77 | 23.37 | **5.05** | 16.37 | 0.094 | 0.168 | 0.078 | 0.136 |
| UrbanDiT | **12.61** | **21.09** | **5.61** | **14.44** | 5.58 | **15.53** | **0.092** | **0.166** | **0.077** | **0.129** |

Table 2: Performance comparison for grid-based forward prediction evaluated using MAE and RMSE. The results are the average prediction errors across all prediction steps. The best result is highlighted in **bold**, and the second-best is indicated with underlining.

| | TaxiBJ | | FlowSH | | TaxiNYC | | CrowdNJ | | PopBJ | |
|---|---|---|---|---|---|---|---|---|---|---|
| **Model** | **MAE** | **RMSE** | **MAE** | **RMSE** | **MAE** | **RMSE** | **MAE** | **RMSE** | **MAE** | **RMSE** |
| CSDI | 36.66 | 75.89 | 15.53 | 34.77 | 19.56 | 69.10 | 0.34 | 0.74 | 0.18 | 0.32 |
| Imputeformer | 37.13 | 77.53 | 17.67 | 38.96 | 20.28 | 49.85 | 0.39 | 0.71 | 0.21 | 0.34 |
| Grin | 41.73 | 92.61 | 22.56 | 47.76 | 22.44 | 58.15 | 0.51 | 0.71 | 0.23 | 0.38 |
| BriTS | 59.94 | 112.34 | 33.74 | 59.10 | 23.39 | 58.47 | 0.50 | 0.70 | 0.54 | 0.75 |
| UrbanDiT (ours) | **8.10** | **12.23** | **5.44** | **10.17** | **4.91** | **12.52** | **0.099** | **0.155** | **0.084** | **0.146** |

Table 3: Performance comparison for spatial extrapolation evaluated using MAE and RMSE. The results represent the average errors across different extrapolation steps.

datasets with different shapes, we train individual models for each dataset. For more flexible models like UniST and PatchTST, we train a single unified model across multiple datasets.

Table 2 provides a comprehensive benchmark for forward prediction on grid-based data. Appendix Table 7 illustrates the results for graph-based data. As observed, traditional deep learning models such as STResNet, ACFM, and MC-STL, do not deliver competitive performance. Similarly, video prediction models, such as MAU, MIM, and SimVP, reveal limitations, suggesting the difference between urban spatio-temporal dynamics and those in conventional video data. UniST demonstrates relatively strong performance, suggesting that training a universal model across different datasets holds potential for improving prediction accuracy. However, time-series forecasting models struggled to capture the complex spatial interactions inherent in urban environments, indicating that precisely modeling these interactions is critical for achieving better results in urban spatio-temporal prediction. Notably, CSDI ranks second in most cases, showing the effectiveness of diffusion-based models in capturing complex patterns within urban spatio-temporal data. Our proposed model, UrbanDiT, delivers the best performance across different datasets using a single unified model, achieving a relative improvement of 11.3%.

We also compare the backward prediction performance of UrbanDiT with the second-best baseline, CSDI, as shown in Appendix Table 6. Notably, CSDI is specifically trained for backward prediction tasks. However, UrbanDiT not only excels in forward prediction but also surpasses specialized

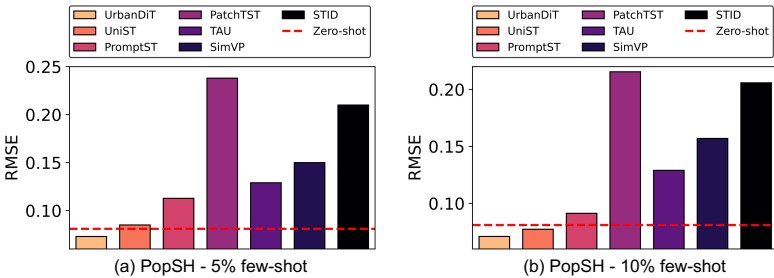

Figure 4: Evaluation of UrbanDiT and baseline models in 5% and 1% few-shot scenarios on the PopSH dataset. The red dashed line indicates UrbanDiT's zero-shot performance

models like CSDI in backward prediction by 30.4%. This result demonstrates UrbanDiT's ability to capture complex spatio-temporal patterns more effectively.

**Temporal Interpolation.** We set the missing ratio to 0.5, meaning that we only know the even-numbered time steps (e.g., 0, 2, 4, ..., 2n), and the model is required to predict the odd-numbered time steps (e.g., 1, 3, 5, ..., 2n-1). Appendix Table 9 demonstrates that UrbanDiT, employing a unified model, outperforms baselines trained separately for different datasets in most cases.

**Spatial Extrapolation.** We evaluate the models' ability to predict missing values in specific spatial regions by masking 50% of of spatial locations across the temporal sequence. The objective is to determine how effectively models extrapolate unobserved spatial information from the remaining visible data. As shown in Table 3, UrbanDiT achieves the best performance in most cases.

**Spatio-Temporal Imputation.** This task assesses the models' capacity to impute missing values across both spatial and temporal dimensions. We randomly mask 50% of positions in the 3D spatio-temporal data, simulating real-world scenarios where urban data may be incomplete due to sensor failures or irregularities in data collection. As shown in Appendix Table 10, UrbanDiT achieves the best performance in most cases.

These results substantiate that UrbanDiT consistently delivers superior performance across diverse tasks and datasets using a single, unified model. This capability positions UrbanDiT as a general-purpose foundation model, enabling practitioners to leverage optimized parameters directly, thereby simplifying deployment and enhancing applicability in urban spatio-temporal applications.

## 4.2 Few-shot and Zero-shot Performance

A key strength of foundation models is their ability to generalize easily. Therefore, we perform experiments in both few-shot and zero-shot scenarios, testing its adaptability to new datasets with little or no additional training. In the *few-shot* scenario, we train UrbanDiT on a small portion of the target dataset—specifically using only 5% and 10% of the available data—and then evaluate its performance on the corresponding test set. This setup challenges the model to generalize well from sparse data. In the *zero-shot* scenario, no data from the target dataset is provided for training. Instead, we directly evaluate UrbanDiT's performance on the target dataset, relying solely on its pretrained knowledge to handle unseen data without any fine-tuning.

Figure 4 demonstrates the few-shot and zero-shot performance of UrbanDiT in comparison to baseline models. In the few-shot setting (with 5% and 1% of the training data), UrbanDiT consistently outperforms baselines, showing its strong ability to learn from minimal data. Even more striking, in the zero-shot scenario, UrbanDiT exhibits exceptional inference capabilities, surpassing nearly all baseline models that had access to training data. This highlights its generalization ability without fine-tuning, reinforcing its effectiveness as an open-world foundation model.

## 4.3 Ablation Studies

**Prompt.** Unified prompt learning is a key design in UrbanDiT. To investigate the contribution of each prompt to the final performance, we conduct ablation studies by systematically removing each type of prompt. Specifically, we identify four types of prompts: $F$ for frequency-domain prompt,

$T$ for time-domain prompt, $S$ for spatial prompt, and $M$ for task-specific prompt. We denote the removal of a prompt as w/o $\{F, T, S, M\}$ and indicate the absence of any prompt as w/o $P$.

Figure 5 presents the results of ablation studies. The findings reveal that removing any single prompt significantly degrades the model's performance. In the absence of prompt design altogether, the model exhibits the poorest performance. Among the four types of prompts, the removal of the frequency-domain prompt has the most pronounced negative impact on the overall performance.

**Inference Steps of Diffusion Models.** We further investigate the effect of inference steps on the performance of diffusion models. The number of inference steps is a critical factor in balancing the model's accuracy and efficiency. Appendix Figure 8 illustrates the performance of the diffusion model across different numbers of inference steps for two datasets, TaxiBJ and TaxiNYC, measured using RMSE. Notably, we observe that around 20 inference steps provide the

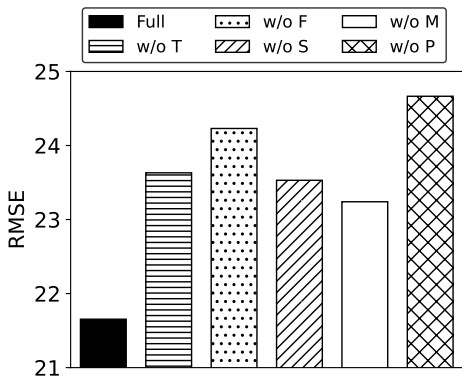

Figure 5: Ablation study on the prompt design using RMSE on the TaxiBJ dataset.

optimal balance between computational efficiency and model performance for both datasets. By setting the diffusion steps to 500 and the inference steps to 20, we achieve a 25x improvement in efficiency compared to the original DDPM [17], without sacrificing accuracy.

## 4.4 Scalability

As a foundation model, it is crucial to understand how model performance evolves as the datasize scale varies across different model sizes. This information is valuable for practitioners to train and fine-tune the foundation model effectively. In Figure 6, we explore the relationship between model performance and datasize scale for three model sizes: UrbanDiT-S (small), UrbanDiT-M (medium), and UrbanDiT-L (large). As observed, all three models demonstrate improved performance as the data size increases. However, when the dataset size increases from 0.8 to 1, the large model, UrbanDiT-L, shows a notably steeper improvement (with a slope of 0.011), compared to the medium (slope of 0.0015) and small models (slope of 0.0019). This pronounced scaling ef-

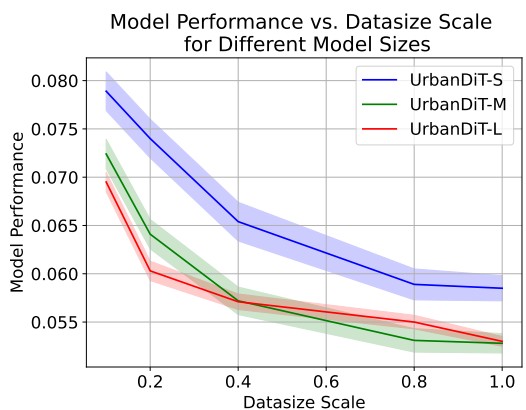

Figure 6: The scalability of UrbanDiT.

fect for the large model indicates its potential to further enhance performance as more data becomes available. These results highlight the promising scalability of UrbanDiT-L, suggesting that it can handle larger datasets and achieve even better outcomes with increased data size.

## 5 Conclusion

In this paper, we present UrbanDiT, an open-world foundation model built on diffusion transformers and a unified prompt learning framework. UrbanDiT enables seamless adaptation to a wide range of urban spatio-temporal tasks across diverse datasets from urban environments. Our extensive experiments highlight the model's exceptional potential in advancing the field of urban spatio-temporal modeling. We believe this work not only pushes the boundaries of urban spatio-temporal modeling but also serves as an inspiration for future research in the rapidly evolving field of foundation models. Nonetheless, UrbanDiT currently focuses on human activity data such as mobility and traffic. To support holistic urban modeling, future work should incorporate environmental variables like air pollution, climate indicators, and microclimate dynamics.

## Acknowledgements

This work was supported by the National Natural Science Foundation of China (No. U24B20180, 62476152). We would also like to express our sincere gratitude to Yu Zheng for the insightful discussions, which were invaluable to this research.

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

Table 4: Basic statistics of grid-based data.

| Dataset | City | Type | Temporal Period | Spatial partition | Interval | Mean | Std |
|---------|------|------|-----------------|-------------------|----------|------|-----|
| FlowSH | Shanghai | Mobility flow | 2016/04/25 - 2016/05/01 | $20 \times 20$ | 15min | 31.935 | 137.926 |
| PopBJ | Beijing | Crowd flow | 2021/10/25 - 2021/11/21 | $28 \times 24$ | One hour | 0.367 | 0.411 |
| TaxiBJ | Beijing | Taxi flow | 2013/06/01 - 2013/10/30 | $32 \times 32$ | Half an hour | 97.543 | 122.174 |
| CrowdNJ | Nanjing | Crowd flow | 2021/02/02 - 2021/03/01 | $20 \times 28$ | One hour | 0.872 | 1.345 |
| TaxiNYC | New York City | Taxi flow | 2015/01/01 - 2015/03/01 | $10 \times 20$ | Half an hour | 38.801 | 103.924 |
| PopSH | Shanghai | Dynamic population | 2014/08/01 - 2014/08/28 | $32 \times 28$ | One hour | 0.175 | 0.212 |

Table 5: Basic statistics of Graph-based data.

| Dataset | City | Type | Temporal Period | Interval | #Nodes | #Edges | Mean | Std |
|---------|------|------|-----------------|----------|--------|--------|------|-----|
| SpeedSH | Shanghai | Traffic speed | 2022/01/27 - 2022/02/27 | 15min | 21099 | 39065 | 7.815 | 4.044 |
| SpeedBJ | Beijing | Traffic speed | 2022/03/05 - 2022/04/05 | 15min | 13675 | 24444 | 6.837 | 3.412 |
| SpeedNJ | Nanjing | Traffic speed | 2022/03/05 - 2022/04/05 | 15min | 13419 | 25100 | 6.699 | 4.253 |

## A Datasets

We provide a detailed overview of the datasets utilized in our study to support future research in the field of urban spatio-temporal modeling. The datasets are categorized into two distinct types: grid-based and graph-based spatio-temporal data. Each type of data reflects different spatial organizations and dynamics, enabling a comprehensive evaluation of model performance across varied urban scenarios.

Grid-based data represent spatial information in a structured, uniform grid layout, where each grid cell corresponds to a specific geographical area. Table 4 outlines the essential details and statistics for the grid-based datasets, including spatial resolution, temporal resolution, temporal period, and the size of each dataset.

Graph-based data, on the other hand, capture urban spatial relationships through a network of nodes and edges, where nodes typically represent points of interest (e.g., intersections or key locations), and edges represent the connections between them (e.g., roads or transit lines). This type of data is well-suited for modeling scenarios that involve irregular spatial structures, such as transportation networks. Table 5 provides a comprehensive summary of the graph-based datasets, including information on the number of nodes, edges, temporal resolution, temporal period, and dataset size.

## B Methodology Details

### B.1 Sequential Format of Input Data

We provide a detailed description of the data unification process for both grid-based and graph-based spatio-temporal data. The key goal is to transform the data into a unified sequential format suitable for the transformer's input.

Grid-based data is structured in a uniform grid layout, typically represented in a three-dimensional form $X_{\text{grid}} \in \mathbb{R}^{T \times H \times W}$ with two spatial dimensions (height $H$ and width $W$) and one temporal dimension $T$. To process this data, we utilize 3D Convolutional Neural Networks (3D CNN), which are widely used for capturing both spatial and temporal dependencies in spatio-temporal tasks. The process is formulated as follows:

$$X' = \text{CONV3D}(X_{\text{grid}}, \text{kernel size} = (p_t, p_s, p_s))$$
$$X_p = \text{RESHAPE}(X', [N])$$

where $N = \frac{T}{p_t} \times \frac{H}{p_s} \times \frac{W}{p_s}$ represents the total number of spatio-temporal partitions, effectively converting the data into a one-dimensional sequence for further processing by the transformer model.

Graph-based data is inherently non-Euclidean, capturing relationships between urban entities (e.g., streets and intersections). The spatial dimension is represented by a graph structure with nodes and edges, and the temporal dimension is still captured as a time series at each node. The graph-based

data can be represented as a tensor $X_{\text{graph}} \in \mathbb{R}^{N \times T}$, where $N$ is the number of nodes in the graph, and $T$ is the number of time steps. To handle the temporal dimension, we first apply a 1D convolutional network (1D CNN) along the time axis to capture local temporal dependencies. Next, to capture spatial relationships, we apply a Graph Convolutional Network (GCN) [55] on the graph structure. For each temporal patch, the GCN aggregates information from neighboring nodes using the graph's adjacency matrix $A \in \mathbb{R}^{N \times N}$. Finally, we reshape the graph-based data into a sequential format. The operations are formulated as follows:

$$X' = \text{CONV1D}(X_{\text{graph}}, \text{kernel size} = p_t)$$
$$X' = \text{GCN}(X', A, W)$$
$$X_p = \text{RESHAPE}(X', [M])$$

where $M$ represents the number of spatio-temporal patches, ensuring that the graph-based data is transformed into a one-dimensional sequence, similar to the grid-based data. This unified sequential representation allows both data types to be processed consistently by the transformer model.

## B.2 Unified Prompt Learning

We provide details of how to obtain the data-driven and task-specific prompts.

**Time-domain patterns.** Suppose the patched spatio-temporal data is denoted as $X \in \mathbb{R}^{T' \times N'}$, where $T' = \frac{T}{p_t}$ and $N' = \frac{H}{p_s} \times \frac{W}{p_s}$. we extract time-domain patterns by applying an attention mechanism along the temporal dimension. This is done independently for each spatial location, allowing us to capture temporal dependencies across different spatial patches as follows:

$$X_t = \text{ATTENTION}(X^T), X^T \in \mathbb{R}^{N' \times T'}, X_t \in \mathbb{R}^{N' \times 1 \times D}$$

where $D$ is the embedding size.

**Frequency-domain patterns.** In our work, we employ four distinct approaches to compute features in the frequency domain, depending on the configuration of the Fast Fourier Transform (FFT) and thresholding mechanisms:

- **Without FFT Threshold**: we directly compute the FFT of the input tensor. The tensor is permuted along the appropriate dimensions, and the real and imaginary components of the FFT are concatenated along the last dimension. This results in a frequency domain representation of the data. It is formulated as follows:

$$X_{\text{FFT}} = \text{FFT}(X),$$
$$X_{\text{freq}} = [\Re(X_{\text{FFT}}), \Im(X_{\text{FFT}})],$$

  where $\Re(X_{\text{FFT}})$ represents the real part of the FFT, and $\Im(X_{\text{FFT}})$ represents the imaginary part.

- **Basic FFT Threshold**: we apply a basic threshold technique by computing the amplitude of the FFT and creating a binary mask. The mask retains frequency components whose amplitude is greater than the mean amplitude, filtering out low-frequency noise and preserving significant frequency components. The process is formulated as follow:

$$X_{\text{FFT}} = \text{FFT}(X),$$
$$A = |X_{\text{FFT}}|, \ \mu_A = \frac{1}{H \times W \times T} \sum A,$$
$$M = \mathbb{I}(A > \mu_A), \ X_{\text{FFT,filtered}} = X_{\text{FFT}}, \odot M,$$
$$X_{\text{freq}} = [\Re(X_{\text{FFT,filtered}}), \Im(X_{\text{FFT,filtered}})].$$

- **Quantile-based FFT Threshold**: We further refine the frequency selection by applying a threshold based on the 80t% of the amplitude distribution. This approach retains the most prominent frequency components, allowing for more flexible filtering compared to the mean-based threshold.

The selection process can be formulated as follows:

$$X_{\text{FFT}} = \text{FFT}(X),$$
$$A = |X_{\text{FFT}}|, \quad q_{80} = \text{Quantile}(A, 0.8),$$
$$M = \mathbb{I}(A > q_{80}), \quad X_{\text{FFT,filtered}} = X_{\text{FFT}} \odot M,$$
$$X_{\text{freq}} = [\Re(X_{\text{FFT,filtered}}), \Im(X_{\text{FFT,filtered}})].$$

- **Top-k Frequency Filtering**: We retain only the top k frequency components (e.g., the first three). We generate a mask to preserve only these dominant components, filtering out the rest. It is formulated as follows:

$$X_{\text{FFT}} = \text{FFT}(X), \quad A = |X_{\text{FFT}}|,$$
$$\text{indices} = \text{argsort}(A, \text{descending})[: k],$$
$$M = \text{mask}(\text{indices}), \quad X_{\text{FFT,filtered}} = X_{\text{FFT}} \odot M,$$
$$X_{\text{freq}} = [\Re(X_{\text{FFT,filtered}}), \Im(X_{\text{FFT,filtered}})].$$

**Spatial patterns.** For the same patched spatio-temporal data $X \in \mathbb{R}^{T' \times N'}$, we extract spatial patterns by applying an attention mechanism along the spatial dimension, independently on each temporal patch. This process allows us to model spatial dependencies within each time patch as follows:

$$X_s = \text{ATTENTION}(X), X \in \mathbb{R}^{T' \times N'}, X_t \in \mathbb{R}^{T' \times 1 \times D}$$

## C   Experiment Details

### C.1   Baselines

- **HA**: History Average is a forecasting method that predicts future values by calculating the mean of historical data from the same time periods.

- **MIM** [44]: This model utilizes the difference in data between consecutive recurring states to address non-stationary characteristics. By stacking multiple MIM blocks, it can capture higher-order non-stationarity in the data.

- **MAU** [6]: The Motion-aware Unit extends the temporal scope of prediction units to seize correlations in motion between frames. It encompasses an attention mechanism and a fusion mechanism, which are integral to video prediction tasks.

- **SimVP** [14]: A simple yet effective video prediction model is entirely based on convolutional neural networks and employs MSE loss as its performance metric, providing a reliable benchmark for comparative studies in video prediction.

- **TAU** [38]: The Temporal Attention Module breaks down temporal attention into two parts: within-frame and between-frames, and employs differential divergence regularization to manage variations across frames.

- **STResNet** [54]: STResNet employs residual neural networks to detect proximity, periodicity, and trends in the temporal data.

- **ACFM** [25]: The Attentive Crowd Flow Machine model forecasts crowd movements by using an attention mechanism to dynamically integrate sequential and cyclical patterns.

- **STGSP** [58]: This model highlights the significance of global and positional temporal data for spatio-temporal forecasting. It incorporates a semantic flow encoder to capture temporal position cues and an attention mechanism to handle multi-scale temporal interactions.

- **MC-STL** [56]: MC-STL utilizes mask-enhanced contrastive learning to efficiently identify spatio-temporal relationships.

- **STNorm** [9]: It introduces two distinct normalization modules: spatial normalization for handling high-frequency elements and temporal normalization for managing local components.

- **STID** [36]: This MLP-based spatio-temporal forecasting model discerns subtleties within the spatial and temporal axes, showcasing its design's efficiency and efficacy.

- **PromptST** [57]: An advanced pre-training and prompt-tuning methodology tailored for spatio-temporal forecasting.

- **UniST** [51]: A versatile urban spatio-temporal prediction model that uses grid-based data. It employs various spatio-temporal masking techniques for pre-training and fine-tuning with spatio-temporal knowledge-based prompts.

- **STGCN** [50]: The Spatio-Temporal Graph Convolutional Network is a deep learning architecture for predicting traffic patterns, harnessing both spatial and temporal correlations. It integrates graph convolutional operations with convolutional sequence learning to capture multi-scale dynamics within traffic networks.

- **GWN** [47]: Graph WaveNet is a technique crafted to overcome the shortcomings of current spatial-temporal graph modeling methods. It introduces a self-adjusting adjacency matrix and utilizes stacked dilated causal convolutions to efficiently capture temporal relationships.

- **MTGNN** [46]: MTGNN is a framework tailored for multivariate time series analysis. It autonomously identifies directional relationships between variables via a graph learning component and incorporates additional information such as variable attributes.

- **GTS** [35]: GTS is an approach that concurrently learns the topology of a graph alongside a Graph Neural Network (GNN) for predicting multiple time series. It models the graph structure using a neural network, allowing for the generation of distinct graph samples, and aims to optimize the average performance across the distribution of graphs.

- **DCRNN** [22]: The Diffusion Convolutional Recurrent Neural Network is a deep learning framework for spatiotemporal prediction. It treats traffic flow as a diffusion phenomenon on a directed graph, securing spatial interdependencies via two-way random walks and temporal interdependencies through an encoder-decoder setup with scheduled sampling.

- **STEP** [37]:Spatial-temporal Graph Neural Network Enhanced by Pre-training is a framework that uses a pre-trained model to enhance spatial-temporal graph neural networks for better forecasting of multivariate time series data.

- **AGCRN** [2]: The AGCRN framework improves upon Graph Convolutional Networks by incorporating two adaptive components: Node Adaptive Parameter Learning and Data Adaptive Graph Generation. This approach effectively captures nuanced spatial and temporal relationships within traffic data, functioning independently of pre-set graph structures.

- **PatchTST** [32]: It employs patching and self-supervised learning techniques for forecasting multivariate time series. By dividing the time series into segments, it captures long-term dependencies and analyzes each data channel separately using a unified network architecture.

- **iTransformer** [27]: This state-of-the-art model for multivariate time series utilizes attention mechanisms and feed-forward neural network layers on inverted dimensions to emphasize the relationships among multiple variables.

- **Time-LLM** [20]: TIME-LLM represents an advanced approach in applying large-scale language models to time series prediction. It employs a reprogramming strategy that adapts LLMs for forecasting tasks without altering the underlying language model architecture.

- **CSDI** [40]: CSDI is explicitly trained for imputation and can exploit correlations between observed values, leading to significant improvements in performance over existing probabilistic imputation methods.

- **Imputeformer** [31]: It introduces a low-rank inductive bias into the Transformer framework to balance strong inductive priors with high model expressivity, making it suitable for a wide range of imputation tasks.

- **Grin** [8]: GRIN introduces a novel graph neural network architecture designed to reconstruct missing data in different channels of a multivariate time series, outperforming state-of-the-art methods in imputation tasks.

- **BriTS** [5]: BRITS is a method for imputing missing values in time series data, utilizing a bidirectional recurrent neural network (RNN) without imposing assumptions on the data's underlying dynamics.

It is worth noting that the baselines, including UniST [51] and PatchTST [32], can also be trained using multiple datasets. In our comparison experiments, we train these models in a unified manner using

the same diverse datasets to ensure a fair comparison. This approach ensures that the performance gains of UrbanDiT are not merely due to dataset diversity, but reflect the model's true advantage.

## C.2 Experiment Configuration

For UrbanDiT-S (small), the model consists of 4 transformer layers with a hidden size of 256. Both the spatial and temporal patch sizes are set to 2, and the number of attention heads is 4. UrbanDiT-M (medium) is composed of 6 transformer layers with a hidden size of 384, maintaining the same spatial and temporal patch sizes of 2, and 6 attention heads. UrbanDiT-L (large) includes 12 transformer layers, a hidden size of 384, spatial and temporal patch sizes of 2, and 12 attention heads. Each memory pool contains 512 embeddings, with the embedding dimension matching the model's hidden size. The learning rate is set to 1e-4, and the maximum number of training epochs is 500, with early stopping applied to prevent overfitting. The batch size is tailored for each dataset to maintain a similar number of training iterations across them.

## C.3 Metrics.

To assess the performance of UrbanDiT in urban spatio-temporal applications, we employ widely recognized evaluation metrics: Root Mean Square Error (RMSE) and Mean Absolute Error (MAE). Given that UrbanDiT operates as a probabilistic model, we conduct 20 inference runs and use the average result for comparison against the ground truth. We apply the same evaluation framework to the probabilistic baselines, ensuring a consistent and fair assessment of all models.

# D Additional Results

## D.1 Results of Multiple Tasks

Table 6 to Table 10 illustrate additional results of multiple tasks.

## D.2 Few-Shot and Zero-Shot Performance

Figure 7 demonstrates UrbanDiT's few-shot and zero-shot capabilities on the TaxiBJ dataset.

## D.3 Ablation Studies

## D.4 Computational Analysis

table 11 provides an overview of model efficiency in terms of overall training time and inference time. While the training time of UrbanDiT is longer than that of the baseline models due to its inclusion of multiple datasets, it is important to note that training separate models for each dataset and summing the total training time results in comparable times between UrbanDiT and the baseline methods. Furthermore, UrbanDiT achieves the best performance across all datasets with a single, unified model, demonstrating its efficiency and effectiveness in delivering superior results without the need for multiple specialized models. This efficiency is crucial for real-world applications, where scalability is key.

Regarding inference latency, UrbanDiT incurs slightly higher costs due to its diffusion-based generative framework, which involves iterative sampling in the denoising process and multiple sampling for probabilistic prediction. However, with Rectified Flow acceleration, inference is significantly faster, notably outperforming CSDI. The resulting latency is reasonable and practically negligible given the substantial performance gains and unified deployment benefits of the model.

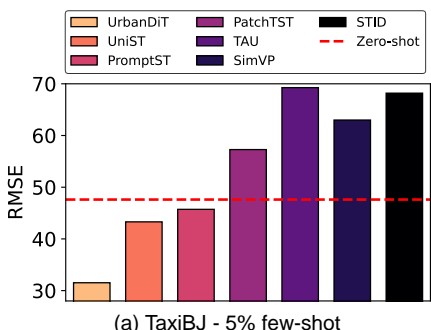 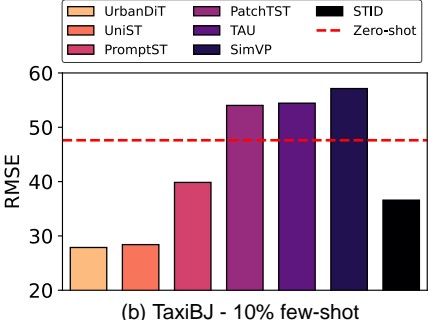

Figure 7: Evaluation of UrbanDiT and baseline models in 5% and 1% few-shot scenarios on the TaxiBJ dataset. The red dashed line indicates UrbanDiT's zero-shot performance

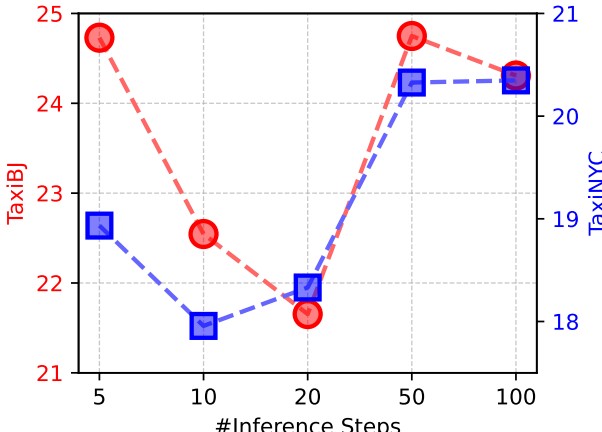

Figure 8: Performance evaluation (RMSE) with varying numbers of inference steps on TaxiBJ and TaxiNYC datasets.

| Model | TaxiBJ | | FlowSH | | TaxiNYC | | CrowdNJ | | PopBJ | |
| --- | --- | --- | --- | --- | --- | --- | --- | --- | --- | --- |
| | MAE | RMSE | MAE | RMSE | MAE | RMSE | MAE | RMSE | MAE | RMSE |
| CSDI | 17.40 | 33.98 | 10.65 | 31.88 | 4.83 | 15.43 | 0.094 | 0.16 | 0.082 | 0.14 |
| UrbanDiT | 11.57 | 20.08 | 5.996 | 14.37 | 4.71 | 15.07 | 0.16 | 0.099 | 0.071 | 0.117 |

Table 6: Performance comparison for grid-based backward prediction evaluated using MAE and RMSE.

|  | SpeedBJ | | SpeedSH | | SpeedNJ | |
|---|---|---|---|---|---|---|
| **Model** | **MAE** | **RMSE** | **MAE** | **RMSE** | **MAE** | **RMSE** |
| HA | 1.35 | 2.13 | 0.92 | 1.46 | 1.94 | 3.01 |
| STGCN | 1.81 | 2.44 | 0.99 | 1.35 | 1.63 | 2.31 |
| CRNN | 1.37 | 1.98 | 0.89 | 1.28 | 1.53 | 2.38 |
| GWN | 1.69 | 2.32 | 0.93 | 1.32 | **1.50** | **2.16** |
| MTGNN | 1.15 | 1.70 | 0.86 | 1.33 | 1.57 | 2.42 |
| AGCRN | 1.66 | 2.29 | 1.14 | 1.56 | 1.77 | 2.46 |
| GTS | 1.76 | 2.36 | 1.31 | 1.74 | 2.04 | 2.68 |
| STEP | 1.45 | 2.04 | 0.93 | 1.32 | 1.58 | 2.42 |
| STID | 1.08 | 1.69 | 0.83 | 1.26 | 1.56 | 2.38 |
| PatchTST | 1.27 | 1.99 | 0.87 | 1.37 | 1.83 | 2.74 |
| PatchTST | 1.55 | 2.44 | 1.08 | 1.70 | 2.19 | 3.34 |
| iTransformer | 1.26 | 1.97 | 0.90 | 1.40 | 1.70 | 2.62 |
| Time-LLM | 1.28 | 2.00 | 0.87 | 1.36 | 1.82 | 2.76 |
| UrbanDiT | **1.02** | **1.66** | **0.78** | **1.20** | 1.51 | 2.30 |

Table 7: Performance comparison of prediction across three graph-based traffic speed datasets.

|  | TaxiBJ | | FlowSH | | TaxiNYC | | CrowdNJ | | PopBJ | |
|---|---|---|---|---|---|---|---|---|---|---|
| **Model** | **MAE** | **RMSE** | **MAE** | **RMSE** | **MAE** | **RMSE** | **MAE** | **RMSE** | **MAE** | **RMSE** |
| CSDI | 11.20 | 18.42 | 5.71 | 13.14 | 3.86 | 11.59 | 0.055 | 0.092 | 0.044 | 0.077 |
| Imputeformer | 11.99 | 19.83 | 6.72 | 15.69 | 5.61 | 16.72 | 0.079 | 0.16 | 0.066 | 0.11 |
| Grin | 13.69 | 23.45 | 9.61 | 26.28 | 8.10 | 21.32 | 0.10 | 0.18 | 0.083 | 0.16 |
| BriTS | 17.57 | 27.63 | 15.24 | 28.40 | 19.41 | 50.25 | 0.19 | 0.28 | 0.16 | 0.25 |
| UrbanDiT (ours) | 9.09 | 14.54 | 4.90 | 10.308 | 4.50 | 11.46 | 0.077 | 0.121 | 0.056 | 0.094 |

Table 8: Performance comparison for temporal interpolation evaluated using MAE and RMSE. The results represent the average errors across different interpolation steps.

|  | TaxiBJ | | FlowSH | | TaxiNYC | | CrowdNJ | | PopBJ | |
|---|---|---|---|---|---|---|---|---|---|---|
| **Model** | **MAE** | **RMSE** | **MAE** | **RMSE** | **MAE** | **RMSE** | **MAE** | **RMSE** | **MAE** | **RMSE** |
| CSDI | 12.29 | 22.07 | 7.94 | 21.86 | 4.33 | 13.09 | 0.071 | 0.12 | 0.055 | 0.094 |
| Imputeformer | 13.65 | 23.18 | 9.22 | 19.97 | 5.95 | 16.36 | 0.093 | 0.16 | 0.069 | 0.12 |
| Grin | 16.83 | 27.61 | 9.70 | 23.52 | 9.15 | 21.43 | 0.16 | 0.30 | 0.096 | 0.18 |
| BriTS | 22.57 | 38.39 | 17.14 | 38.82 | 19.93 | 50.47 | 0.26 | 0.41 | 0.18 | 0.29 |
| UrbanDiT (ours) | 9.38 | 15.19 | 5.03 | 11.52 | 4.62 | 12.16 | 0.083 | 0.13 | 0.061 | 0.101 |

Table 9: Performance comparison for temporal imputation evaluated using MAE and RMSE. The results represent the average errors across different imputation steps.

|  | TaxiBJ | | FlowSH | | TaxiNYC | | CrowdNJ | | PopBJ | |
|---|---|---|---|---|---|---|---|---|---|---|
| **Model** | **MAE** | **RMSE** | **MAE** | **RMSE** | **MAE** | **RMSE** | **MAE** | **RMSE** | **MAE** | **RMSE** |
| CSDI | 7.92 | 12.42 | 4.28 | 8.62 | 3.86 | 11.54 | 0.057 | 0.091 | 0.046 | 0.083 |
| Imputeformer | 9.70 | 13.80 | 5.50 | 10.30 | 4.79 | 15.35 | 0.076 | 0.12 | 0.061 | 0.11 |
| Grin | 11.96 | 19.62 | 9.21 | 19.68 | 9.62 | 20.77 | 0.11 | 0.19 | 0.080 | 0.14 |
| BriTS | 13.99 | 23.53 | 17.95 | 38.57 | 19.17 | 50.15 | 0.21 | 0.44 | 0.13 | 0.19 |
| UrbanDiT (ours) | 7.83 | 12.13 | 5.07 | 9.79 | 3.63 | 11.44 | 0.057 | 0.090 | 0.049 | 0.092 |

Table 10: Performance comparison for grid-based spatio-temporal imputation evaluated using MAE and RMSE. The results represent the average prediction errors across different prediction steps.

Table 11: Model Training and Inference Times

| Model | Train Time All | Inference Time |
|---|---:|---:|
| STGCN | 17 min | 2 s |
| DCRNN | 77 min | 8 s |
| GWN | 16 min | 1 s |
| MTGNN | 14 min | 0.8 s |
| AGCRN | 21 min | 2 s |
| GTS | 126 min | 17 s |
| STEP | 177 min | 27 s |
| STResNet | 5.7 min | 0.6 s |
| ACTM | 56 min | 0.9 s |
| STNorm | 46 min | 5 s |
| STGCP | 8 min | 4 s |
| MC-STL | 31 min | 7 s |
| MAU | 82 min | 13 s |
| MIM | 84 min | 14 s |
| TAU | 22 min | 6 s |
| PromptST | 45 min | 9 s |
| Imputeformer | 28 min | 6 s |
| BriTS | 82 min | 10 s |
| Grin | 17 min | 2 s |
| UniST | 5 h | 19 s |
| STID | 10 min | 5 s |
| PatchTST | 33 min | 5 s |
| iTransformer | 23 min | 6 s |
| Time-LLM | 6 h (multiple datasets) | 5 min |
| CSDI | 5.5 h | 38 min |
| UrbanDiT | 4 h (multiple datasets) | 57 s |

