# OpenReview forum: "Diffusion Transformers as Open-World Spatiotemporal Foundation Models"
_NeurIPS.cc/2025/Conference — NeurIPS 2025 poster_

### Official Review · Reviewer_NF7R · 2025-06-25

**Clarity:** 4
**Significance:** 3
**Originality:** 4
**Rating:** 5
**Confidence:** 5

**Summary:**

This paper presents UrbanDiT, a foundation model designed for open-world urban spatio-temporal learning. Built on a diffusion transformer backbone, UrbanDiT unifies diverse urban datasets, including both grid-based and graph-based data, into a sequential format and supports a wide range of tasks such as bi-directional prediction, temporal interpolation, spatial extrapolation, and spatio-temporal imputation. Its core innovation lies in a unified prompt learning mechanism, incorporating both data-driven prompts and task-specific prompts. UrbanDiT demonstrates state-of-the-art performance across numerous datasets and tasks, including zero-shot and few-shot generalization.

**Questions:**

1.	The prompt ablation study is informative. Could the authors additionally evaluate prompt effectiveness under few-shot conditions? Specifically, do different types of prompts remain beneficial when training data is scarce?

2.	See weakness

**Ethical Concerns:**

["NO or VERY MINOR ethics concerns only"]

**Final Justification:**

My concerns have been resolved, and I recommend accepting the paper.

**Limitations:**

The paper briefly mentions that UrbanDiT focuses on human activity data and does not yet model environmental factors.

**Quality:**

3

**Strengths And Weaknesses:**

**Strengths**

1.	This paper addresses an important and practical challenge in urban spatio-temporal modeling: the fragmentation of models across tasks and scenarios. By proposing a unified perspective, the work advances the goal of general-purpose urban AI systems.

2.	The proposed method presents solid technical innovations. The unified data representation enables seamless integration of grid-based and graph-based data, while the unified prompt learning framework is novel and effectively adapts to heterogeneous urban data and task types.

3.	The empirical evaluation is thorough and convincing, covering a wide range of cities, data modalities (grid and graph), and key urban spatio-temporal tasks including prediction, interpolation, extrapolation, and imputation.

4.	The model demonstrates strong open-world capabilities, including zero-shot and few-shot generalization, which are crucial for real-world deployment in unseen cities or under limited data availability.

5.	The paper is clearly written and well-structured, making the methodology and results easy to understand and follow.

**Weaknesses**

1. The scope of the current model is limited to human-related spatio-temporal data (e.g., mobility, traffic), without incorporating environmental or multi-modal factors such as weather, air quality, or climate variables.

2. While the unified prompt mechanism is effective, its interpretability remains somewhat implicit; it would be beneficial to visualize or analyze how different prompt types contribute across tasks or domains.

---

> ### Author Rebuttal · Authors · 2025-07-30
>
> Dear Reviewer NF7R,
>
> We sincerely thank you for the constructive and insightful feedback. We are encouraged that you acknowledged the technical contributions of our work and its thorough experimental validation. We would like to address the concerns you raised as follows.
>
> **W1. The scope of the current model is limited to human-related spatio-temporal data (e.g., mobility, traffic), without incorporating environmental or multi-modal factors such as weather, air quality, or climate variables.**
>
> **Response.** We appreciate the reviewer’s comment. UrbanDiT focuses on urban spatio-temporal dynamics, which are inherently tied to human activities. Urban systems are largely shaped by human-driven behaviors, such as mobility and traffic, making them central to the scope of our model. UrbanDiT is designed to capture the spatio-temporal patterns most directly influenced by human actions within urban environments.
>
> The environmental and climate factors you mentioned are part of broader, more external spatio-temporal systems, often referred to as natural systems. These systems are indeed important, and incorporating them is a natural direction for future research. However, for this study, we have focused primarily on human-related data, as it is more directly tied to the dynamics of urban systems.
>
> We did attempt to integrate climate spatio-temporal data with urban data in our experiments, but the results did not yield significant improvements. This outcome makes sense, as the dynamics governing urban and natural systems (e.g., climate) are fundamentally distinct. Nevertheless, we recognize the value of exploring the coupling between these two domains in future work, and this remains an important area of ongoing research.
>
>
> **W2. While the unified prompt mechanism is effective, its interpretability remains somewhat implicit; it would be beneficial to visualize or analyze how different prompt types contribute across tasks or domains.**
>
> **Response.** Thanks for your valuable suggestion! To enhance the interpretability, we will include visualizations in the final version to show the spatio-temporal patterns stored in the memory pool ($\underline{\text{Section 4.5}}$). Specifically, we will provide the following:
>
> - We will show the embeddings of the memory units both at the initial state and after optimization. These visualizations will reveal how the model’s memory evolves during training, allowing us to understand the process of how the memory pools adapt over time.
> - We will also visualize the specific spatio-temporal patterns memorized by each unit.
>
> Since we are unable to provide images in this response, we summarize our findings as follows. From the start state to the final optimized state, the embeddings gradually diverge in different directions. This suggests that, throughout the optimization process, the memory pools progressively store and encapsulate more personalized information. Additionally, we find that the memorized patterns revealed through the prompt tool exhibit notable consistency across different urban scenarios. However, the attention weight distributions associated with the memory units for different datasets manifest significant dissimilarities. This indicates that the model’s focus is dynamic and responsive, adjusting according to the specific characteristics of the input data.
>
>
> **Q1. The prompt ablation study is informative. Could the authors additionally evaluate prompt effectiveness under few-shot conditions? Specifically, do different types of prompts remain beneficial when training data is scarce?**
>
> **Response.** Thank you for this important question. Investigating why the model performs well in few-shot and even zero-shot conditions is crucial for understanding its robustness. To address this, we evaluated the effectiveness of prompts under few-shot settings and visualized the patterns memorized by each unit in these conditions. We found that, even with limited or no training data, the model is capable of extracting useful prompts that are highly relevant to the input samples, which effectively guide the learning process. We will include these findings and visualizations in the revised version of the paper ($\underline{\text{Section 4.5}}$) to further highlight this key aspect of the model's performance.
>
> Thank you again for your positive ratings and constructive feedback. We are happy to address any remaining questions.
>
> Sincerely,
>
> The Authors

---

> > ### Comment · Reviewer_NF7R · 2025-08-04
> >
> > Thank you for addressing my questions. My concerns have been resolved, and I recommend accepting the paper.
> >
> > This paper is compatible with multiple spatiotemporal data formats, including grids and graphs, and performs well across diverse tasks, indicating that it could serve as a foundation model for spatiotemporal applications. The observed zero-shot generalization properties further support the real-world deployability of the method. In short, the paper delivers a general-purpose approach that addresses long-standing scalability and structural issues in spatiotemporal modeling. It is well-positioned to influence both academic research and practical deployment in this area.

---

### Official Review · Reviewer_XMSo · 2025-06-29

**Clarity:** 3
**Significance:** 4
**Originality:** 3
**Rating:** 5
**Confidence:** 5

**Summary:**

The paper proposes UrbanDIT, which is a foundation model for urban prediction tasks, especially those dynamic ones such as traffic flows and taxi demand. The model incorporates three major designs, including a unification process of data and tasks, in which all types of data are converted into sequential data, easy to be trained using transformers; unified prompt learning, in which data-driven prompt and task-specific prompt are designed, to enhance the performance of different datasets and tasks; and difussion transformer as backbone network model.

The model is tested in a comprehensive set of experiments including multiple study areas and datasets, making the effectiveness of the model apparent. In fact, some of the performance lifting is eye-catching, such as table 3.

In an overall manner I am supportive of this paper, a very solid and promising contribution to urban foundation models.

**Questions:**

See the questions in weaknesses.

**Ethical Concerns:**

["NO or VERY MINOR ethics concerns only"]

**Final Justification:**

I've read the discussion and I am aware of different opinions from other reviewers. I recognise that this is a solid contribution for an important sub-domain of foundation models. Therefore I would like to keep my rating.

**Limitations:**

yes

**Quality:**

4

**Strengths And Weaknesses:**

Strengths: 1) an important application domain of machine learning and foundation models; 2) a smart and novel design of the model using the previously mentioned three major design choices; 3) experiements are credible and comprehensive, showing the very good performance of the model.

Weaknesses: 1) the justifications of many design choices are lacking, such as why unified prompt learning is needed, and some very concrete real-world examples would be helpful, e.g., maybe a key in a memory pool is linked to morning rush hours? 2) the discussion around other inference tasks in urban analysis is lacking, e.g. how this model can benefit the analysis of land use, house price etc.

---

> ### Author Rebuttal · Authors · 2025-07-30
>
> Dear Reviewer XMSo,
>
> We also greatly appreciate your constructive suggestions, which will help enhance the clarity and depth of our manuscript. In response, we have addressed the points raise, providing clearer justifications and a broader discussion on the model’s applicability. Below is our detailed response to your comments.
>
> **W1. The justifications of many design choices are lacking, such as why unified prompt learning is needed, and some very concrete real-world examples would be helpful, e.g., maybe a key in a memory pool is linked to morning rush hours?**
>
> **Response.** We appreciate the reviewer’s comment. Regarding the justification for unified prompt learning, we emphasize the following key points.
>
> - Unified prompt learning is essential because it faciliates UrbanDiT to seamlessly handle diverse spatio-temporal data and tasks within a single model. This flexibility is particularly important in urban settings, where cities, scenarios, and tasks can vary significantly. Direct joint training across such heterogeneous data could lead to contradictions and inefficiencies. The prompts serve as crucial signals, guiding the model’s learning process. They allow the model to adapt to various tasks and data types, making it possible for these diverse elements to complement and benefit from each other. This is achieved through semantic alignment, which is facilitated by decomposing the data into its spatio-temporal dimensions.
> - As for the example of the memory pools, we have conducted an analysis where different memory keys store varying spatio-temporal patterns. Our findings show that memory units effectively capture specific patterns, such as traffic flow during peak hours. In the revised version, we will visualize these results. While we cannot provide images here, we summarize our findings as follows: the memory embeddings evolve during optimization, progressively storing more personalized information. Importantly, the memorized patterns demonstrate consistency across different urban scenarios. However, attention weights vary across datasets, indicating that the model dynamically adjusts its focus based on the specific characteristics of the input data.
>
>
> We thank you for your suggestion to improve the clarity and explainability of our approach. We will include a more detailed analysis of the technical design ($\underline{\text{Section 3.3}}$) and provide an in-depth exploration of the prompt learning framework ($\underline{\text{Section 4.5}}$) in the revised version.
>
>
> **W2. The discussion around other inference tasks in urban analysis is lacking, e.g. how this model can benefit the analysis of land use, house price etc.**
>
> **Response.** We appreciate your suggestion and agree that a broader discussion on potential inference tasks in urban analysis would be valuable. UrbanDiT focuses on modeling urban spatio-temporal dynamics, which are closely tied to human daily activities. This capability offers significant benefits for urban analysis tasks such as land use prediction and house price forecasting.
>
> Mobility patterns are crucial indicators of how land is being used and will evolve. By modeling the movement of people across different time scales and regions, UrbanDiT can identify areas experiencing increasing commercial or residential activity, as well as shifts in land use.  Moreover, urban spatio-temporal dynamics play a key role in influencing house prices. For example, areas with improved transportation connectivity often experience changes in housing demand, which directly impacts property values. UrbanDiT provides a predictive advantage in identifying emerging trends in housing markets.
>
> In summary, UrbanDiT’s ability to model urban spatio-temporal dynamics enables it to generate actionable insights for a variety of applications, offering concrete benefits in urban planning and real estate analysis. We will incorporate this discussion into the revised manuscript, particularly in the $\underline{\text{discussion and conclusion}}$ section.
>
> Thank you again for your constructive suggestions. We are happy to address any remaining questions.
>
> Sincerely,
>
> The Authors

---

> > ### Comment · Reviewer_XMSo · 2025-08-02
> >
> > Thanks for the response. I will keep my rating.

---

### Official Review · Reviewer_EsKL · 2025-07-01

**Clarity:** 2
**Significance:** 3
**Originality:** 2
**Rating:** 3
**Confidence:** 5

**Summary:**

UrbanDiT pioneers a diffusion transformer (DiT)-based foundation model for unified urban spatio-temporal learning. It addresses the fragmentation in existing approaches by integrating diverse data types into a sequential format via spatio-temporal patching. The core innovation lies in its prompt learning framework: data-driven prompts generated from memory pools guide adaptation to heterogeneous urban data, while task-specific prompts enable a single model to handle five distinct tasks—bidirectional prediction, temporal interpolation, spatial extrapolation, and spatio-temporal imputation.

**Questions:**

1. During model training, does a single task require simultaneous use of both graph-based and grid-based data?
2.  How do embeddings of graph-based and grid-based data achieve semantic alignment in the spatio-temporal domain?

**Ethical Concerns:**

["NO or VERY MINOR ethics concerns only"]

**Final Justification:**

+ The authors addressed specific technical questions regarding the design and workflow of the proposed model in the rebuttal phase.
+ Overall, the paper has some contributions, especially in terms of building a scalable system.
- The paper's general idea and overall framework is quite similar to existing work namely UniST. The new contributions appear to be somewhat incremental for a new paper in NeurIPS.
- The authors emphasized in the rebuttal the enhanced capabilities of their proposed model such as scalability and generalizability. However, the technical innovations behind such achievements are inadequately explained, leading to the suspicion that the advancement is mostly the result of engineering tweaks.

I would like to keep the score, but would not be upset if the paper is accepted.

**Limitations:**

Yes

**Quality:**

2

**Strengths And Weaknesses:**

Strengths：
1. Unified Architecture: Seamlessly processes grid/graph data and multi-task learning, eliminating task-specific models.
2. Efficient Inference: 20-step diffusion reduces inference time by 25 times vs. standard DDPM.

Weaknesses:
1. Stringent Data Requirements: The model mandates both grid and graph-structured inputs for unified training , restricting deployment in tasks lacking either data modality.

2. Limited Spatio-Temporal Data Diversity: Experiments focus on common urban activities, omitting emerging data types like mobile check-ins, bike-sharing patterns, or social media mobility traces, which may reveal finer urban dynamics.

3. Limited technical contributions. The proposed model does not provide much new insights or design ideas. Instead, it is a simple ensemble of existing techniques. For example, the memory pool design largely follows recent works such as UniST.  The model simply uses diffusion transformers as the major building blocks with no modification.

---

> ### Author Rebuttal · Authors · 2025-07-30
>
> Dear Reviewer EsKL,
>
> We sincerely thank you for your thoughtful review and constructive feedback. We appreciate your recognition of the strengths of our model. In response to the concerns you raised, we have addressed the issues regarding data requirements,  data diversity, and the technical contributions of UrbanDiT. Below is our detailed response to each of your comments.
>
> **W1. Stringent Data Requirements: The model mandates both grid and graph-structured inputs for unified training , restricting deployment in tasks lacking either data modality.**
>
> **Response.** We appreciate the reviewer’s comment. However, we would like to clarify that **UrbanDiT does not require the simultaneous use of both grid and graph-structured inputs**. Instead, it is designed to handle either grid or graph data independently, providing flexibility for practitioners to select suitable data type based on the specific task.
>
> This flexibility is, in fact, one of our key contributions. The data unification strategy enables the model to seamlessly process diverse data types, whether grid-based  or graph-based. This addresses the concern of restricting deployment in tasks lacking either data modality, as UrbanDiT can function effectively with either type of data.
>
> We will clarify this point in $\underline{\text{Section 3.2 ("Unification of Data and Tasks")}}$ in the revised version to prevent potential misunderstandings.
>
>
> **W2. Limited Spatio-Temporal Data Diversity: Experiments focus on common urban activities, omitting emerging data types like mobile check-ins, bike-sharing patterns, or social media mobility traces, which may reveal finer urban dynamics.**
>
> **Response.** We appreciate the reviewer’s valuable suggestion. In response, we have incorporated additional urban scenarios, such as bike-sharing data (BikeNYC) and mobile check-in data (MobileSH), into our experiments. These additions contribute to enhancing data diversity and provide a more comprehensive view of urban spatiotemporal dynamics. The following tables show the updated results, which will be incorporated into $\underline{\text{Table 2}}$ in the final version of our paper.
>
>
> | Model        | MobileSH MAE | MobileSH RMSE | BikeNYC MAE | BikeNYC RMSE |
> |--------------|--------------|---------------|-------------|--------------|
> | HA           | 0.100        | 0.165         | 7.17        | 15.68        |
> | ARIMA        | 0.112        | 0.197         | 13.63       | 25.01        |
> | PatchTST     | 0.062        | 0.108         | 5.30        | 12.33        |
> | iTransformer | 0.045        | 0.072         | 4.50        | 9.86         |
> | STResNet     | 0.102        | 0.138         | 3.94        | 7.18         |
> | ATFM         | 0.055        | 0.078         | 3.09        | 5.99         |
> | STNorm       | 0.042        | 0.065         | 3.03        | 6.47         |
> | STGSP        | 0.040        | 0.057         | 7.38        | 14.20        |
> | TAU          | 0.044        | 0.063         | 2.89        | 5.98         |
> | PromptST     | 0.043        | 0.069         | 2.76        | 5.84         |
> | MAU          | 0.081        | 0.125         | 2.95        | 6.12         |
> | MIM          | 0.079        | 0.126         | 2.89        | 6.36         |
> | STID         | 0.040        | 0.062         | 2.71        | 5.70         |
> | UniST        | 0.039        | 0.055         | 2.56        | 5.50         |
> | CSDI         | 0.045        | 0.072         | 2.26        | 5.22         |
> | UrbanDiT     | 0.034        | 0.052         | 2.04        | 5.01         |
>
>
>
> Regarding individual mobility traces from social media, we currently do not support this data type due to the irregular time intervals and discrete location tokens. We recognize the potential value of incorporating such data in the future and are actively exploring ways to unify discrete and numerical data types within our framework. However, the absence of this feature does not hinder the model’s ability to effectively capture and analyze the numerical spatio-temporal dynamics of urban systems that are central to our focus.
>
>
>
>
> **W3. Limited technical contributions. The proposed model does not provide much new insights or design ideas. Instead, it is a simple ensemble of existing techniques. For example, the memory pool design largely follows recent works such as UniST. The model simply uses diffusion transformers as the major building blocks with no modification.**
>
> **Response.**  We respectfully disagree with the reviewer’s characterization of UrbanDiT as lacking novel insights or design ideas. We highlight the following key innovations that distinguish our work from UniST and represent significant advancements in spatio-temporal modeling.
>
> **First, UrbanDiT addresses a critical gap in spatio-temporal modeling  by creating a true “one-for-all” foundation model.** Unlike UniST, which is primarily focused on prediction tasks and limited to grid-based data, UrbanDiT is the first model to unify spatio-temporal modeling across four key dimensions: data format, city, domain, and task, within a single model trained from scratch. This unification enables UrbanDiT to handle diverse scenarios in open-world settings, offering greater flexibility and scalability compared to UniST.
>
>
> **Second, we introduce key methodological innovations to the spatio-temporal domain.** While UrbanDiT builds on general-purpose frameworks like the diffusion transformer architecture, our contribution lies in developing a foundational model based on this architecture, not simply applying it directly. We introduce novel designs to address the unique challenges of spatio-temporal modeling. Our data unification strategy transforms diverse data formats into a unified representation, enabling UrbanDiT to seamlessly process different types of urban data. Additionally, we propose a unified prompt learning framework that generates both data-driven and task-specific prompts, allowing joint learning across heterogeneous data and tasks without conflicts. Unlike NLP or CV, urban spatiotemporal modeling faces greater challenges due to the highly distinct patterns across different scenarios and cities. **The key innovation of our framework is ensuring that these diverse scenarios and tasks complement and benefit from each other during the joint training.**
>
>
>
> **Lastly, an obvious distinctive feature of UrbanDiT is its performance gain over UniST.**  UrbanDiT is capable of capturing broader spatio-temporal patterns across different data formats, cities, domains, and tasks, making it a more versatile foundation model.
>
>
>
> In summary, UrbanDiT introduces substantial methodological advancements that set it apart from UniST. Our focus on cross-format, cross-task, and cross-region generalization represents a significant step forward in spatio-temporal modeling. We believe these contributions establish UrbanDiT as a more flexible, scalable, and generalizable model.
>
>
> **Q1. During model training, does a single task require simultaneous use of both graph-based and grid-based data?**
>
>
> **Response.**  To clarify, UrbanDiT does not require the simultaneous use of both graph-based and grid-based data as inputs for a single task.  Instead, UrbanDiT is designed to handle either graph-based or grid-based data independently, depending on the task at hand. This flexibility is a key feature of the model, allowing UrbanDiT to adapt to diverse urban spatio-temporal tasks without requiring both data types simultaneously.
>
> **Q2. How do embeddings of graph-based and grid-based data achieve semantic alignment in the spatio-temporal domain?**
>
> **Response.** Although UrbanDiT does not require the simultaneous input of both graph-based and grid-based data, it supports flexible integration of both, which inherently implies a form of semantic alignment.
>
> Before explaining the alignment, we would like to clarify that graph and grid data have different spatial organizations. Graph-based data represents spatial relationships in a network structure, while grid-based data organizes spatial data in a regular, structured grid. Despite these differences, both share key spatial characteristics, such as neighborhood relationships. In the temporal dimension, alignment is more natural, as both data types share common temporal features, like temporal closeness and periodicity.
>
> To achieve semantic alignment, we generate spatially-related and temporally-related prompts that align the underlying semantics of these data types. Specifically, for different input samples, the model adaptively retrieves the most relevant prompt, ensuring that the alignment between spatial and temporal semantics is maintained.  This adaptive retrieval process facilitates a unified learning process across heterogeneous data.
>
> Thank you once again for your thoughtful and constructive feedback. We believe the additional experiments and clarifications regarding the technical contributions address your primary concerns. Should you find our response and the updated results satisfactory, we would be grateful if you would consider raising your rating. We are happy to address any further questions or provide additional clarifications as needed.
>
> Sincerely,
>
> The Authors

---

> > ### Comment · Reviewer_EsKL · 2025-08-03
> > **Further discussion**
> >
> > I would like to thank the authors for their detailed responses.
> >
> > The authors clearly illustrated "what" the proposed model can do, particularly integrating more heterogeneous data types and handling diverse tasks. But I'm more interested in learning what the key technical innovations are to enable the advancements in a nut shell. It appears to me that the current model achieves the goal through incremental modifications here and there, with some engineering tweaks. This observation aligns with the comment of reviewer DoyW (but I do not agree with the rating of 5 given such observations).
> >
> > Therefore, I'm not fully convinced that the paper's contribution merits a better score. I still feel that my current rating is appropriate.

---

> > > ### Comment · Reviewer_EsKL · 2025-08-03
> > >
> > > Also I would like to hear other reviewers' opinions on why the paper's technical innovations merit a rating of 5.

---

> > > > ### Comment · Reviewer_DoyW · 2025-08-07
> > > >
> > > > I think ST foundation model is an important problem that has not been fully addressed in previous works. I also believe that designing methods within foundation models, such as handling data scalability, instructing tasks, and facilitating joint training, to create a system that performs well in open-world scenarios, is a valuable contribution to this field.

---

> ### Author Response · Authors · 2025-08-03
> **Clarifying the technical innovations**
>
> **Thank you again for your continued engagement and thoughtful feedback.**
>
> We appreciate your effort to differentiate *what* the model can do from *how* it achieves it through technical innovation. To clarify our contributions from a methodological perspective, we note that recent breakthroughs in fields like time series, human mobility, cellular network, and Earth system modeling [1-4]  consistently highlight three axes of innovation: **(i)** unified architectures across modalities, **(ii)** scalable training across heterogeneous data, and **(iii)** flexible task adaptation. These axes precisely frame UrbanDiT’s core design.
>
>
> We argue that in domain-specific foundation modeling, technical innovation often lies not in inventing new architectural primitives, but in **designing a scalable, extensible system that successfully solves the grand unification challenge of the target domain**. This view is supported by seminal works across scientific fields [1–4], where innovation arises from integrating powerful components into cohesive frameworks that generalize broadly across real-world scenarios. For instance, AlphaFold3 [5] represents a major leap beyond AlphaFold [6] not because of increased architectural complexity, but because of a shift in modeling paradigm: it simplifies the network backbone while expanding to richer multi-modal inputs and adopts diffusion models. This trend underscores that in domain-specific foundation models, **system-level unification and data-driven scalability are central to technical innovation**.
>
>
> While UrbanDiT builds on diffusion and Transformer backbones, just as *Sundial*, *Aurora* and *UniTraj* do, it introduces a holistic framework for **cross-format, cross-task, and cross-domain** urban modeling. By this standard, it makes substantive contributions as the first foundation model tailored to the complexity of open-world urban spatio-temporal learning.
>
> We hope this clarifies our position and would greatly appreciate your reconsideration.
>
>
> [1]Bodnar, C., Bruinsma, W. P., Lucic, A., Stanley, M., Allen, A., Brandstetter, J., ... & Perdikaris, P. (2025). A foundation model for the Earth system. Nature, 1-8.
>
> [2] Liu, Y., Qin, G., Shi, Z., Chen, Z., Yang, C., Huang, X., ... & Long, M. Sundial: A Family of Highly Capable Time Series Foundation Models. In Forty-second International Conference on Machine Learning.
>
> [3] Zhu, Y., Yu, J. J., Zhao, X., Wei, X., & Liang, Y. (2024). Unitraj: Learning a universal trajectory foundation model from billion-scale worldwide traces. arXiv preprint arXiv:2411.03859.
>
>
> [4] Chai, H., Zhang, S., Qi, X., Qiu, B., & Li, Y. (2025). UoMo: A Universal Model of Mobile Traffic Forecasting for Wireless Network Optimization. In Proceedings of the 31st ACM SIGKDD Conference on Knowledge Discovery and Data Mining
>
> [5] Abramson, J., Adler, J., Dunger, J., Evans, R., Green, T., Pritzel, A., ... & Jumper, J. M. (2024). Accurate structure prediction of biomolecular interactions with AlphaFold 3. Nature, 630(8016), 493-500.
>
> [6] Jumper, J., Evans, R., Pritzel, A., Green, T., Figurnov, M., Ronneberger, O., ... & Hassabis, D. (2021). Highly accurate protein structure prediction with AlphaFold. nature, 596(7873), 583-589.

---

> > ### Comment · Reviewer_EsKL · 2025-08-04
> >
> > Thank you for further input on this.
> >
> > I agree that "designing a scalable, extensible system that successfully solves the grand unification challenge of the target domain" is a valid contribution. The key questions are (1) what are the specific technical challenges in building such a system in this paper, that existing framework and techniques (e.g., UniST) cannot be generalized to address? (2) what are the specific, non-trivial technical efforts the authors had to make to address the challenges.
> >
> > I'm still not sure what specific non-trivial technical efforts the author have made for "integrating powerful components into cohesive frameworks that generalize broadly across real-world scenarios". From the descriptions above, the efforts are more on the engineering side to me. But I invite the authors to provide more information.

---

> > > ### Author Response · Authors · 2025-08-05
> > > **Response to Reviewer’s Follow-up Comment**
> > >
> > > Thank you once again for your thoughtful follow-up! We truly appreciate your willingness to engage more deeply with our contributions. Below, we address your two central questions in detail.
> > >
> > >
> > > **(1) What are the specific technical challenges in building such a system, that existing frameworks (e.g., UniST) cannot be generalized to address?**
> > >
> > > **Response.** We agree this is a crucial point, and would like to clarify three key technical challenges.
> > >
> > > The first challenge lies in unifying heterogeneous spatial structures, as grids (e.g., air quality) and graphs (e.g., traffic flows) differ not only in format but also in how spatial dependencies are defined. Regular grid adjacency contrasts with the irregular topology of graphs, making it difficult to model them jointly. Learning a unified representation requires capturing and aligning shared spatio-temporal dynamics that transcend format-specific designs, which is not supported by prior models like UniST.
> > >
> > >
> > > The second challenge concerns the unification of diverse spatio-temporal tasks. Real-world urban systems involve a wide variety of tasks with different supervision structures, data availability, and target objectives. Designing a model that can generalize flexibly across such tasks under partial, asynchronous, or missing observations is not trivial.
> > >
> > >
> > > The third challenge involves modeling uncertainty. Spatio-temporal systems are inherently chaotic, where small input changes can cause large output deviations. Deterministic models like UniST reduce this variability to a single prediction, making them brittle under noise and unable to capture the full distribution of possible futures.
> > >
> > >
> > > **(2) What are the specific, non-trivial technical efforts the authors had to make to address the challenges?**
> > >
> > > **Response.** UrbanDiT addresses these challenges through a set of carefully designed and mutually reinforcing components.
> > >
> > > The first major effort is the joint design of a unified spatio-temporal patching mechanism and a spatial prompt learning module, which together enable modality-agnostic modeling.   We construct a patch-based tokenization strategy that transforms both formats into sequential representations while preserving spatial semantics, using node and edge features for graphs, and position-aware patches for grids. On top of this shared input space, we introduce spatial prompts that guide attention in a topology-agnostic way. Rather than relying on fixed adjacency, these prompts learn latent spatial priors.  This separation between structure-aware encoding and adaptive interaction modeling allows UrbanDiT to generalize effectively across heterogeneous data formats.
> > >
> > > The second effort is the development of a dual-prompt mechanism that enables task-conditioned control and generalization. One prompt encodes the task type, while the other is dynamically retrieved from a learned memory pool. This design allows UrbanDiT to handle diverse spatio-temporal tasks under a unified generative framework without modifying model architecture or performing task-specific fine-tuning.
> > >
> > >
> > > The third effort is the integration of a diffusion-based model to enable **generative** spatio-temporal capabilities. This is not a superficial adoption of diffusion; instead, rather, we carefully designed the aforementioned data unification and prompt mechanisms to make it effective in spatio-temporal settings. Such generative capability is essential for advancing toward reasoning-enhanced spatio-temporal modeling, and we are actively exploring this promising extension of the UrbanDiT framework.
> > >
> > > **In summary**, These designs are non-trivial because they require reconciling incompatible assumptions across modalities (e.g., spatial adjacency), building a unified token space for Transformers, and coordinating conditional generation across dynamic task and data settings. All components are tightly coupled under a unified modeling principle, rather than assembled independently. We hope this clarifies that **UrbanDiT is a system-level solution that reimagines how unified generative modeling can empower the urban spatio-temporal domain**.
> > >
> > >
> > > We are sincerely grateful for your thoughtful questions and continued engagement, as they have pushed us to reflect more deeply on the core challenges and contributions of our work. We truly appreciate your constructive feedback. Please let us know if any further clarification would be helpful.

---

### Official Review · Reviewer_DoyW · 2025-07-02

**Clarity:** 4
**Significance:** 3
**Originality:** 3
**Rating:** 5
**Confidence:** 4

**Summary:**

The paper proposes UrbanDiT, a foundation model for open-world urban spatio-temporal learning. It unifies diverse data types (grid and graph) into a sequential format and supports multiple tasks through a prompt learning framework combining data-driven and task-specific prompts. Built on diffusion transformers, UrbanDiT handles various flow prediction tasks in a unified way. Extensive experiments across multiple cities and domains show that UrbanDiT achieves state-of-the-art performance and strong zero-shot generalization. The model further demonstrates robust few-shot performance, effective masking strategies for task control, and scalability across model sizes and dataset scales.

**Questions:**

Please see in Weakness

**Ethical Concerns:**

["NO or VERY MINOR ethics concerns only"]

**Final Justification:**

I think my concerns have been addressed. What I appreciate is that the paper tackles an important and meaningful problem. Therefore, I will maintain my positive score.

**Limitations:**

yes

**Paper Formatting Concerns:**

No formating concerns

**Quality:**

4

**Strengths And Weaknesses:**

Strengths:

1. Unified framework for heterogeneous data: The model handles both grid-based and graph-based spatiotemporal data in a unified sequential format, which allows it to be applied across a variety of urban datasets with different spatial structures.

2. Consistent formulation of diverse tasks: By representing multiple spatiotemporal tasks through masking strategies within a diffusion transformer framework, the model offers a coherent way to address forward/backward prediction, interpolation, extrapolation, and imputation.

3. Comprehensive empirical evaluation: The paper provides extensive experiments across multiple cities and task types. Results show that the model performs competitively compared to a range of strong baselines, including both task-specific and general-purpose methods.

Weaknesses:

1. The model largely builds upon existing ideas such as diffusion transformers and prompt-based conditioning. While the integration is systematic, the contribution is more in engineering design than in fundamental innovation.

2. The use of multiple memory pools (temporal, spatial, frequency) increases model complexity, but the paper provides limited discussion on how to balance effectiveness with interpretability or potential overfitting.

3. Lack of computational analysis: The paper does not report training time, inference latency, or memory usage, which are important for assessing the practicality of scaling the model in real-world urban systems.

---

> ### Author Rebuttal · Authors · 2025-07-31
>
> Dear Reviewer DoyW,
>
> We appreciate the positive feedback regarding the strengths of our approach, particularly the unified framework for heterogeneous data, the consistent formulation of diverse tasks, and the comprehensive empirical evaluation across multiple cities and domains.  We have carefully addressed the concerns raised in the review, and the following is a point-by-point response to each of them.
>
> **W1. The model largely builds upon existing ideas such as diffusion transformers and prompt-based conditioning. While the integration is systematic, the contribution is more in engineering design than in fundamental innovation.**
>
> **Response.** We respectfully argue that our contributions go beyond engineering design and offer meaningful technical innovations for the spatio-temporal modeling community.
>
>
> **First, UrbanDiT tackles a critical and underexplored challenge: building a true “one-for-all” urban spatio-temporal foundation model.** To the best of our knowledge, it is the first model to unify diverse data formats while spanning multiple application domains and task types, all within a single model trained from scratch. Unlike previous universal or foundation models, which mainly focus on prediction tasks and are often limited to grid-based spatio-temporal data, UrbanDiT achieves a level of generalization across formats, tasks, scenarios, and regions that has not been realized by existing approaches. We believe this represents a significant step forward in the development of open-world spatio-temporal modeling.
>
> **Second, we introduce key methodological innovations to the spatio-temporal domain.** The DiT+Prompt structure is a general-purpose framework that has been successfully applied in various fields [1,2,3]. Our contribution lies in building a foundational model based on this general architecture, but not simply applying it directly. Instead, we make novel designs to address the unique challenges of spatio-temporal modeling. Our data unification strategy transforms diverse data formats into a unified representation, enabling UrbanDiT to seamlessly process different types of urban data. Additionally, we introduce a unified prompt learning framework that generates both data-driven and task-specific prompts, allowing joint learning across heterogeneous data and tasks without conflicts. Unlike NLP or CV, urban spatiotemporal modeling faces greater challenges due to the highly distinct patterns across different scenarios and cities. The key innovation of our framework is ensuring that these diverse data sources benefit from each other during joint training, making UrbanDiT highly adaptable for a wide range of urban tasks.
>
>
>
> **Last, UrbanDiT exhibits emergent and scalable generalization abilities.** As shown in our experiments, UrbanDiT demonstrates emergent zero-shot generalization across cities and domains. These capabilities indicate that UrbanDiT captures underlying spatio-temporal regularities beyond specific tasks, which we view as a step toward a universal spatio-temporal foundation model.
>
>
>
> [1] Timedit: General-purpose diffusion transformers for time series foundation model. arXiv preprint.
> [2] Text2PDE: Latent Diffusion Models for Accessible Physics Simulation, ICLR 2025
> [3] Diffusion on Language Model Encodings for Protein Sequence Generation, ICML 2025
>
> **W2. The use of multiple memory pools (temporal, spatial, frequency) increases model complexity, but the paper provides limited discussion on how to balance effectiveness with interpretability or potential overfitting.**
>
>
> **Response.** We appreciate the reviewer’s concern about the potential trade-off between model complexity, interpretability, and the risk of overfitting when using multiple memory pools in UrbanDiT. We address these points as follows:
>
> **First, each memory pool is designed to capture distinct, interpretable factors** that reflect the core elements of urban spatio-temporal dynamics. These include temporal patterns (e.g., daily/weekly cycles), spatial correlations (e.g., neighborhood effects), and frequency-based latent periodicities.  This modular approach improves model interpretability and enables targeted analysis of each component's contribution. As shown in our ablation studies (Fig. 6, Section 4.3), **each memory pool uniquely contributes to the model's performance**. Removing any pool leads to significant degradation, indicating that each captures valuable, non-redundant information.
>
> To enhance interpretability, we will include visualizations in the final version ($\underline{\text{Section 4.5}}$) showing the spatio-temporal patterns stored in the memory pools. Since we cannot provide images in this response, we summarize our findings here. From the initial to the final state, the embeddings in the memory pools diverge, reflecting the storage of increasingly personalized information. The memorized patterns show consistency across urban scenarios, while attention weights vary across datasets, indicating the model dynamically adjusts its focus based on input data characteristics.
>
> **Second, regarding overfitting, UrbanDiT mitigates this risk through its robust generalization across domains, cities, and tasks**. Its training paradigm alternates between datasets and tasks, acting as a natural regularizer and preventing overfitting to any single dataset. Strong zero-shot and few-shot performance (Section 4.3, Fig. 5) demonstrates its ability to generalize to unseen cities, outperforming models trained on target-domain data. This empirical evidence supports the claim that UrbanDiT captures robust, transferable patterns rather than memorizing dataset-specific details.
>
>
> **W3. Lack of computational analysis: The paper does not report training time, inference latency, or memory usage, which are important for assessing the practicality of scaling the model in real-world urban systems.**
>
> **Response.** Thanks for this valuable suggestion. We have now included a computational analysis. The following table provides an overview of model efficiency in terms of overall training time and inference time.
>
> While the training time of UrbanDiT is longer than that of the baseline models due to its inclusion of multiple datasets, it is important to note that training separate models for each dataset and summing the total training time results in comparable times between UrbanDiT and the baseline methods. Furthermore, UrbanDiT achieves the best performance across all datasets with a single, unified model, demonstrating its efficiency and effectiveness in delivering superior results without the need for multiple specialized models. This efficiency is crucial for real-world applications, where scalability is key.
>
> Regarding inference latency, UrbanDiT incurs slightly higher costs due to its diffusion-based generative framework, which involves iterative sampling in the denoising process and multiple sampling for probabilistic prediction.  However, with Rectified Flow acceleration, inference is significantly faster, notably outperforming CSDI. The resulting latency is reasonable and practically negligible given the substantial performance gains and unified deployment benefits of the model.
>
>
> | Model        | Train Time All | Inference Time |
> |--------------|----------------|----------------|
> | STGCN        | 17 min         | 2 s            |
> | DCRNN        | 77 min         | 8 s            |
> | GWN          | 16 min         | 1 s            |
> | MTGNN        | 14 min         | 0.8 s          |
> | AGCRN        | 21 min         | 2 s            |
> | GTS          | 126 min        | 17 s           |
> | STEP         | 177 min        | 27 s           |
> | STResNet     | 5.7 min        | 0.6 s          |
> | ACTM         | 56 min         | 0.9 s          |
> | STNorm       | 46 min         | 5 s            |
> | STGCP        | 8 min          | 4 s            |
> | MC-STL       | 31 min         | 7 s            |
> | MAU          | 82 min         | 13 s           |
> | MIM          | 84 min         | 14 s           |
> | TAU          | 22 min         | 6 s            |
> | PromptST     | 45 min         | 9 s            |
> | Imputeformer | 28 min         | 6 s            |
> | BriTS        | 82 min         | 10 s           |
> | Grin         | 17 min         | 2 s            |
> | UniST        | 5 h            | 19 s           |
> | STID         | 10 min         | 5 s            |
> | PatchTST     | 33 min         | 5 s            |
> | iTransformer | 23 min         | 6 s            |
> | Time-LLM     | 6 h (multiple datasets) | 5 min |
> | CSDI         | 5.5 h          | 38 min         |
> | UrbanDiT      | 4 h (multiple datasets) | 57 s |
>
>
> We will include this analysis in the $\underline{\text{Appendix D}}$  in the revised version to provide a comprehensive view of UrbanDiT’s scalability and real-world practicality.
>
> Thank you again for your constructive feedback. We are happy to address any remaining questions.
>
>
> Sincerely,
>
> The Authors

---

> > ### Comment · Reviewer_DoyW · 2025-08-04
> >
> > Thank you for the authors’ detailed response. I think my concerns have been addressed. What I appreciate is that the paper tackles an important and meaningful problem. Therefore, I will maintain my positive score.

---

### Author Response · Authors · 2025-08-09
**Summary of Rebuttal**

We sincerely thank all the reviewers for their insightful reviews and valuable comments, which are instructive for us to improve our paper further.

The reviewers generally expressed positive opinions about our paper, noting that the proposed method demonstrates "strong zero-shot generalization," tackles "an important and meaningful problem," and includes "solid technical innovations." The reviewers acknowledged that our work is "well-positioned to influence both academic research and practical deployment in this area" and considered it "a valuable contribution to the field." Additionally, they highlighted that the paper is "well-written," with "extensive" and "comprehensive" experiments.

The reviewers also raised insightful and constructive concerns. Below is a summary of the key points from our rebuttal, highlighting how we have addressed the concerns raised during the review process.


-  **Clarification of the technical contribution：** We addressed concerns raised from Reviewer DoyW and Reviewer EsKL by further clarifying the technical contributions of UrbanDiT, particularly its unified framework and scalability across tasks, data formats, and regions.
- **Addition datasets：** In response to Reviewer EsKL’s suggestion, we incorporated mobile check-ins and bike-sharing data to further demonstrate the model’s ability to handle diverse data types and enhance data diversity.
- **Computational analysis：** Following Reviewer DoyW's suggestion, we included a detailed computational analysis to showcase the model’s practicality in scaling for real-world urban systems.
- **Clarification of model designs：** We clarified the integration of different data formats and emphasized the importance of unified prompt learning. We also added experiments to provide greater explainability of the prompt mechanism.
- **Future work discussion：** Based on discussions with Reviewer EsKL and input from Reviewer XMSo, we expanded the discussion on potential future applications of UrbanDiT, including land use and house price forecasting, and explored future extensions toward reasoning-enhanced spatio-temporal modeling.


We are confident that the rebuttal process have significantly strengthened the manuscript, making UrbanDiT a more robust, clear, and impactful contribution to the field. We sincerely appreciate the reviewers' helpful suggestions.


Sincerely,

The Authors

---

### Decision · Program_Chairs · 2025-09-17

**Decision:**

Accept (poster)

**Comment:**

This paper proposes UrbanDiT, a diffusion transformer–based foundation model for open-world urban spatio-temporal learning. The model unifies heterogeneous urban data into a sequential representation and introduces a prompt learning framework combining data-driven and task-specific prompts, enabling a single model to handle diverse tasks, such as prediction, interpolation, extrapolation, and imputation. Extensive experiments across multiple cities demonstrate strong performance, scalability, and promising zero-shot/few-shot generalization.

**Strengths:**
1. It addresses an important and practical problem in building general-purpose models for urban spatio-temporal tasks.
2. The unified data representation and prompt mechanism are well-motivated, making the approach flexible across tasks and modalities.
3. The paper provides comprehensive empirical validation, with consistent improvements over baselines, strong generalization ability, and clear presentation.

**Weaknesses:**
1. The model builds upon existing techniques, such as diffusion transformers, prompt learning and memory pooling, raising concerns that contributions are more engineering integration than methodological novelty.
2. The requirement of both grid and graph modalities may limit deployment, and experiments focus mainly on mobility/traffic datasets, without broader multimodal or environmental data.
3. The paper does not thoroughly report training/inference cost, memory efficiency, or scalability trade-offs, which are important for real-world applications.
4. The roles of different prompts and memory pools could be better justified and illustrated with concrete urban examples.

Overall, despite these weaknesses, I agree with most reviewers that the paper makes a solid and meaningful contribution to urban AI, with a well-designed framework and strong empirical results. Therefore, I recommend acceptance. Meanwhile, the final version should include further discussion on technical novelty, efficiency, and broader applicability as discussed during the rebuttal.